# Revealing 3D microanatomical structures of unlabeled thick cancer tissues using holotomography and virtual H&E staining

Juyeon Park[1,2,14], Su-Jin Shin [3,14], Geon Kim[1,2], Hyungjoo Cho[4], Dongmin Ryu [4], Daewoong Ahn[4], Ji Eun Heo[5], Jean R. Clemenceau [6], Isabel Barnfather[6], Minji Kim [6], Inyeop Jang[6], Ji-Youn Sung [6,7], Jeong Hwan Park [6,8], Hyun-seok Min [4], Kwang Suk Lee[9], Nam Hoon Cho[10], Tae Hyun Hwang [6,11,12,13] ✉ & YongKeun Park [1,2,4] ✉

In histopathology, acquiring subcellular-level three-dimensional (3D) tissue structures efficiently and without damaging the tissues during serial sectioning and staining remains a formidable challenge. We address this by integrating holotomography with deep learning and creating 3D virtual hematoxylin and eosin (H&E) images from label-free thick cancer tissues. This method involves measuring the tissues' 3D refractive index (RI) distribution using holotomography, followed by processing with a deep learning-based image translation framework to produce virtual H&E staining in 3D. Applied to colon cancer tissues up to 50 μm thick—far surpassing conventional slide thickness—this technique provides direct methodological validation through chemical H&E staining. It reveals quantitative 3D microanatomical structures of colon cancer with subcellular resolution. Further validation of our method's repeatability and scalability is demonstrated on gastric cancer samples across different institutional settings. This innovative 3D virtual H&E staining method enhances histopathological efficiency and reliability, marking a significant advancement in extending histopathology to the 3D realm and offering substantial potential for cancer research and diagnostics.

Histopathology, the meticulous microscopic examination of tissue samples, is fundamental in identifying structural and cellular alterations within diseased tissues. For centuries, the application of hematoxylin and eosin (H&E) staining has been the cornerstone of this process, enhancing the visibility of cellular and tissue architectures under bright-field microscopy. Such procedures have been instrumental in diagnosing, grading, and classifying diseases by providing valuable insights into microscopic structural alterations. Nonetheless, this conventional method is primarily confined to two-dimensional (2D) analysis, which significantly limits its capacity to reveal the intricate three-dimensional (3D) architecture inherent in biological tissues[1]. The constraints posed by bright-field microscopy and H&E

staining become particularly evident when attempting to analyze thick 3D tissue samples, thereby restricting our understanding of 3D histopathological structures. Recent advancements underscore the indispensable role of 3D analysis in histopathology, offering insights into critical features like vascular and neural structures[2], tissue heterogeneity[3,4], and nuanced aspects of cancer grading[5,6]. Current methods for achieving 3D histopathological analysis, however, entail labor-intensive sample preparation processes, including extensive sectioning, staining, or optical clearing. Such procedures are not only demanding in terms of time and resources but also prone to introducing artifacts during the serial sectioning and staining steps—artifacts that are irreversible once they occur.

In response to the considerable demands of 3D histopathology, significant strides have been made towards eliminating traditional staining processes. Notably, quantitative phase imaging (QPI) has emerged as a pioneering label-free technique that quantitatively captures the optical phase delay in specimens, offering a non-invasive alternative for examining histopathological samples[7,8]. With the advantages of label-free imaging capabilities and quantitative measurements, QPI has become an invaluable tool for investigating histopathological specimens, including structures of diverse cancers[9–18], Alzheimer's disease[19], and Crohn's disease[20]. While the studies predominantly focused on 2D specimens, recent advancements in 3D QPI techniques are now being applied to various histopathological specimens[21–24].

Recently, deep learning has been integrated with QPI, enhancing its ability to discern subcellular structures through the innovative concept of virtual staining[25–27]. Virtual staining involves an image-to-image translation framework, using deep learning to convert label-free images to the desired stained images. This approach has been applied to diverse histopathological slides and target stains. For instance, unsupervised deep learning has successfully generated immunohistochemistry-stained images from phase images of kidney tissue slides[28]. Additionally, supervised deep learning enabled virtual staining of nuclei and F-actin using the label-free phase images of kidney tissue slides[29]. Furthermore, Phasestain has generated H&E, John's, and Masson's trichrome stained images from label-free phase images of skin, kidney, and liver tissue, respectively[30]. These studies have demonstrated the potential of virtual staining to make standard histopathology more efficient by eliminating the staining procedures and replacing them with computational staining. However, it is important to note that these research efforts have primarily focused on 2D applications. A recent study has demonstrated that the integration of the 3D QPI technique with virtual staining has enabled 3D virtual H&E staining of tissue[24], revealing the 3D structure of tumor margins through virtual H&E images. However, due to the absence of ground truth images for comparison, the validation of the proposed framework heavily relies on indirect approaches.

In this study, we confront two predominant challenges in histopathology: the limitation to two-dimensional analysis and the intensive resources required by traditional H&E staining procedures. Our innovative strategy leverages holotomography combined with deep learning to bypass conventional staining, facilitating the generation of three-dimensional H&E-stained images without direct staining (Fig. 1a, b). Holotomography, a 3D QPI technique, reconstructs the label-free 3D refractive index (RI) distribution of the sample with high spatial resolution and optical sectioning capabilities. To realize the holotomography, we utilize the deconvolution phase microscopy which involves the deconvolution of intensity images using optical transfer functions to capture the 3D RI distribution of specimen without external staining[31] (Fig. 1c). A deep learning technique for image-to-image translation is integrated with the holotomography to generate virtual H&E stained images from label-free RI images in 3D (Fig. 1d). The framework includes training and testing phase. During the training phase, the neural network learns to map between the RI images and H&E stained images obtained from a conventional 4 μm-thick H&E-stained tissue slide. Once the training is completed, the optimized neural network is directly applied to the 3D RI images acquired from thick, label-free cancer tissue slides, resulting in the 3D virtual H&E stained images. Specifically, we employ a supervised learning approach, utilizing direct ground truth images to validate the accuracy of virtual H&E images, ensuring the reliability and credibility of our framework (Fig. 1e).

## Results

### Network training

To train the network, we prepared the paired training dataset from a 4 μm-thick, H&E-stained colon cancer tissue slide (Fig. 2a). Initially, we utilized holotomography to capture both the 3D RI images and 2D single-channel BF (scBF) images. To integrate cellular details from various axial positions into a single focal plane, we applied an all-in-focus technique to the 3D RI images[32] (See "Methods"). Subsequently, we imaged the same slide using a whole slide scanner (WSS) to acquire H&E-stained BF images. Then, image registration was conducted to align the BF images obtained from WSS and scBF images acquired from holotomography. This alignment resulted in a paired dataset of BF and RI images since scBF and RI images are inherently aligned due to being captured at identical positions. The registration procedures were based on the spatial transform network[33] (See "Methods"). Further refinement involved cropping these images into patches of 1024 × 1024 pixels and applying the same registration techniques to each patch to enhance accuracy. Patches with a Pearson's correlation coefficient below 0.65 between the scBF and BF images were excluded to maintain data quality. The final curated dataset comprised 2538 paired patches, which were then divided into two subsets: 1996 patches for training and 542 patches for validation

For the architecture of the neural network, we employed a conditional generative adversarial network to train the model (Fig. 1e). The network consists of a generator and discriminator, which are adversarially learning to map RI images to BF images. Scalable neural architecture search (SCNAS) has been used as the generator, an optimized architecture for 3D medical image datasets[34] (Supplementary Fig. 2a). The discriminator consists of five convolutional layers (Supplementary Fig. 2b). While the generator creates the virtual H&E stained images from input RI images, the discriminator's role is to distinguish the created image from the ground truth images when provided with input images.

To assess the performance of the trained networks, we compared virtual H&E images and corresponding ground truth images. First, we obtained a wide-field, all-in-focused RI image from a different region of the same slide used for training (Fig. 2b), which revealed various glandular structures (Fig. 2bi–iv). Subsequently, the wide-field RI image was cropped into 1024 × 1024-pixel patches with a 50% overlap and we fed the patches into the trained network. The resultant virtual H&E images were stitched back to reconstruct the original wide-field image using the ImageJ plugin[35] (Fig. 2c). For a direct quantitative comparison between the virtual H&E images and the ground truth images, we obtained the WSS image of the corresponding region (Fig. 2d). The results demonstrate that the trained network successfully predicted most of the histological and anatomical features. To further validate its performance, we compared various histological features, including subnuclear structures and signet ring cells, between the virtual and chemical H&E images (Fig. 2e). The virtual H&E images accurately captured key features, such as subnuclear details, the skewed nuclear position characteristic of signet ring cells, and mucinous regions, which were clearly distinguishable from the vacant background or surrounding cytoplasm. Moreover, circular structures of normal mucosa, with nuclei aligned along the perimeter, and the pinkish aggregated structures of necrosis were successfully reproduced by our trained network.

In our quantitative analysis, the structural similarity index measure (SSIM) was employed to evaluate the fidelity of virtual H&E images against their chemically stained counterparts across selected regions of interest. To account for the resolution discrepancies between holotomography-derived images and those obtained from the WSS, we first normalized the spatial frequency ranges of both datasets, ensuring a fair comparison of intensity levels before proceeding with SSIM calculations. This adjustment allowed for a meaningful comparison that accurately reflects the structural preservation in the virtual staining process. We then extended this evaluation to the entire field by calculating SSIM values for individual 1024 × 1024-pixel patches across the slides. The SSIM values ranged from 0.75 to 0.83, with an average of 0.78, indicating a consistent

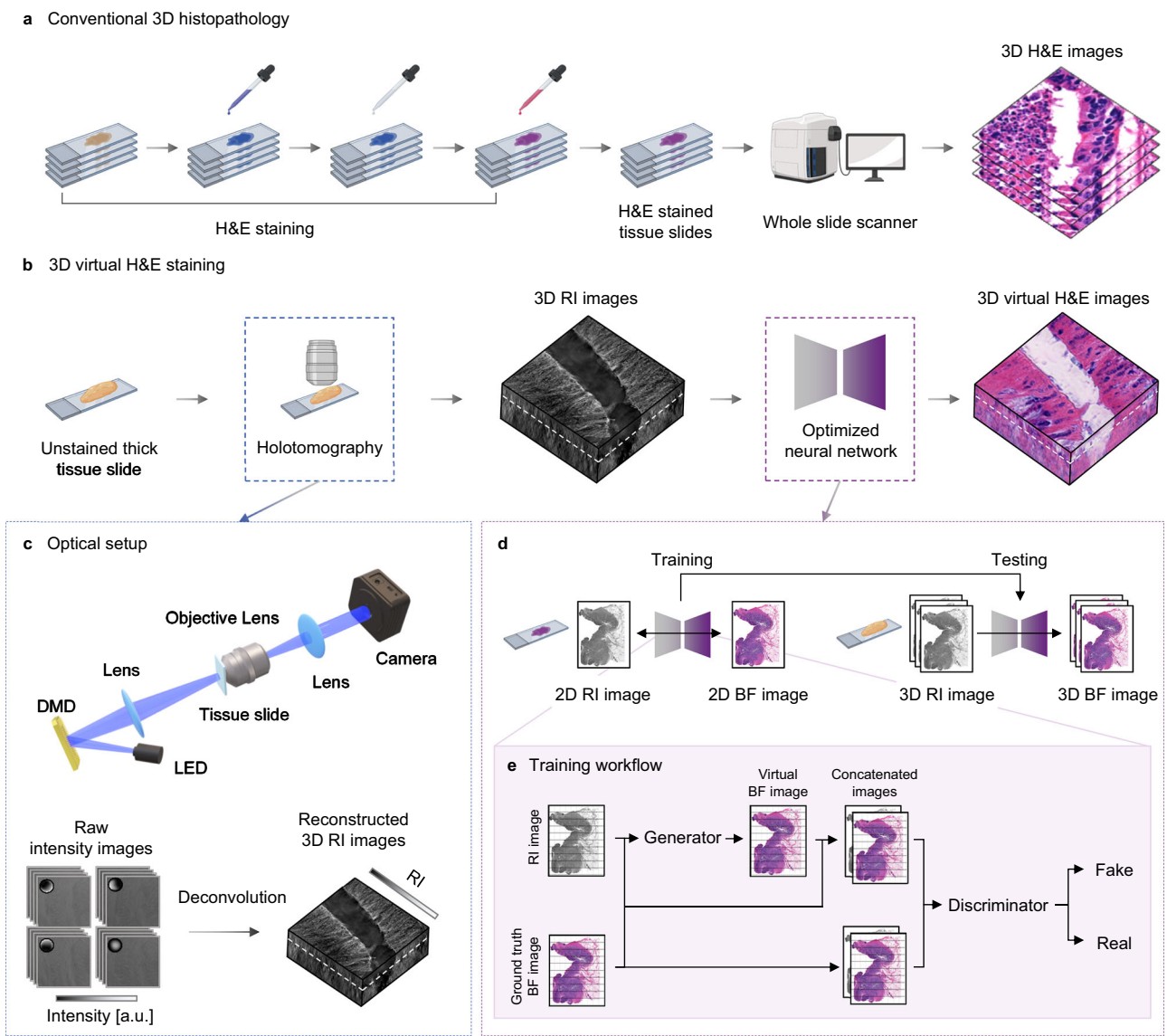

**Fig. 1 | Overview of the proposed framework. a** Procedures for conventional 3D histopathology including multiple sectioning, H&E staining, and imaging under BF microscopy. **b** The proposed framework of 3D virtual H&E staining. Integration of holotomography and deep learning enabled label-free 3D virtual H&E staining of thick cancer tissue slides. **c** Optical setup for holotomography and reconstruction procedures. Raw intensity images are deconvolved with the optical transfer functions, reconstructing 3D RI images. **d** Training and testing phases for 3D virtual staining. The neural network is trained to map between RI and BF images using the conventional 4 μm-thick H&E-stained tissue slides during the training phase. The trained neural network is directly applied to the 3D RI images obtained from label-free, thick tissue slides. **e** A supervised learning workflow employed during the training phase. Figures 1a, b, d were created with BioRender.com.

level of similarity between the virtual and chemical H&E images across the dataset (Fig. 2f).

While SSIM values are a prevalent metric for evaluating the performance of virtual staining techniques, it's critical to recognize their limitations. Sole reliance on SSIM can sometimes lead to counterintuitive findings, as this measure may not fully capture the nuances of staining fidelity or the biological relevance of the imaged structures[36,37]. Consequently, we directed our evaluation efforts towards analyzing the virtual staining quality through the lens of H&E staining's essential attributes. Given that the primary purpose of H&E staining is to facilitate the visualization of nuclei and cytoplasm, we conducted a comparative analysis of nuclei detection in both virtual and chemical H&E images (Fig. 2g–j). Utilizing HoVer-Net[38], we segmented nuclei from virtual and chemical H&E images. Overlaying the segmentation maps revealed a predominance of overlapped pixels (yellow) across the wide-field images (Fig. 2h). To quantitatively assess the degree of overlap,

Jaccard indices were computed for each 1024 × 1024-pixel patch, yielding a mean value of 0.5 (Fig. 2g).

In addition to pixel-level comparative analysis, we compared essential histological parameters such as nucleus area and quantity. We calculated the average number and size of nuclei in each patch from both segmentation maps. Statistical comparison between the results showed no significant difference between the values computed from virtual and chemical H&E images (Figs. 2i, j). These results suggest that quantitative nuclei identification, a primary objective of H&E staining, yields consistent results regardless of whether they are derived from virtual or chemical H&E images.

From a model architecture perspective, we evaluated several image translation networks for the virtual staining task, including U-Net[39], conditional diffusion[40], and pix2pix[41] (Supplementary Fig. 1 and Supplementary Table 1). Quantitatively, both U-Net and our algorithm achieved the highest SSIM and PSNR values, with U-Net exhibiting a slight advantage in these metrics. However, qualitative

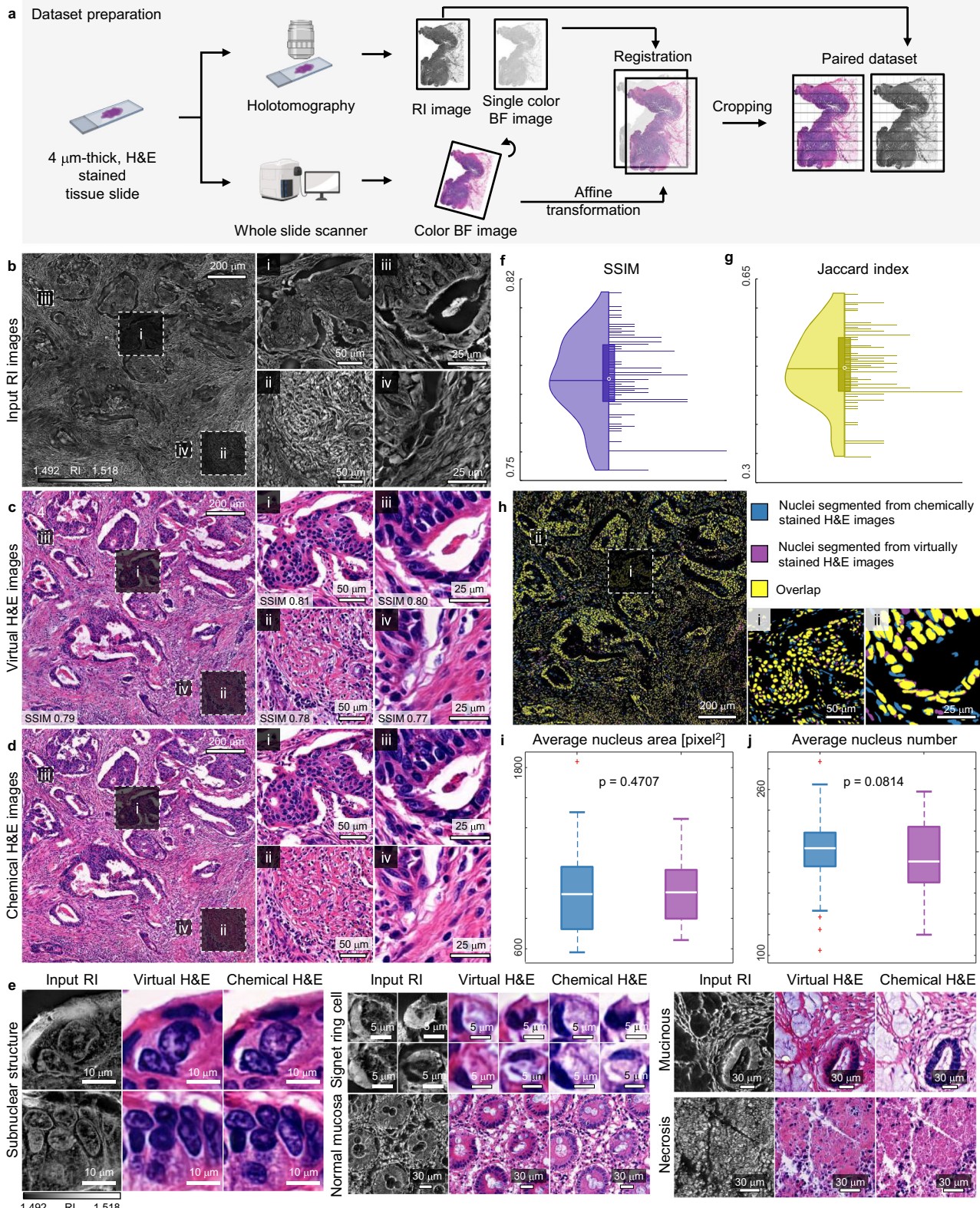

assessments are equally critical in histopathology, where expert interpretations of morphological features are paramount. Accordingly, detailed comparisons of U-Net- and our algorithm-generated images (Supplementary Figs. 1c−e) showed that U-Net tends to produce more blurred outputs, while our method more accurately preserves intricate tissue structures. Considering both quantitative metrics and qualitative evaluations, we conclude that our algorithm is better suited for the goals of virtual staining.

## 3D virtual H&E staining of thick colon cancer tissue slides

To create 3D virtual H&E images of label-free thick cancer tissue, we sourced an additional colon cancer slide from a different patient, capturing its 3D RI images. This slide, prepared at a thickness of 10 µm −2.5 times thicker than standard histopathology slides−was imaged without staining. Employing holotomography, we acquired wide-field 3D RI images covering ~1.2 × 1.2 mm regions of the slide. These images vividly rendered anatomical structures characteristic of colon cancer,

**Fig. 2 | Validations of the trained network with a 4 m-thick, H&E-stained colon cancer tissue slide. a** Dataset preparation steps before network training. **b** A wide-field RI image obtained from a 4 μm-thick, H&E-stained colon cancer slide and its detailed images. Results are from a single experiment. **c** A wide-field virtual H&E image generated by the trained neural network and its detailed images. Results are from a single experiment. **d** A ground truth chemical H&E image obtained using a WSS and its detailed images. Results are from a single experiment. **e** Comparison of virtual and chemical H&E images across various pathological features. Results are from a single experiment. **f** Distribution of SSIM values computed from 100 cropped patches. In the box plot, the white circle at the center of the box represents the median, the ends of the box represent the first and third quartiles, and the whicker spans a 1.5 interquartile range from the ends. **g** Distribution of Jaccard index values computed from 100 cropped patches of nucleus segmentation results. In each box plot, the white circle at the center of the box represents the median, the ends of the box represent the first and third quartiles, and the whicker spans a 1.5 interquartile range from the ends. **h** An overlapped nuclei segmentation map obtained from virtual (blue) and chemical (purple) H&E images. Results are from a single experiment. **i, j** Distribution of average nucleus area (**i**) and average number of nuclei (**j**) at each cropped patch obtained from virtual (blue) and chemical (purple) H&E images. The distributions were derived from a total of 100 patches. Paired, two-sided Student's t-test was used to calculate $P$ values. In each box plot, the white line represents the median, the ends of the box represent the first and third quartiles, and the whicker spans a 1.5 interquartile range from the ends. Figure 2a was created with BioRender.com.

such as glands and lumens (Fig. 3b). A deeper exploration of these structures was achieved by zooming into the 3D RI representations, revealing the lumens' vacant regions and the intricate glandular formations around them with clear axial detail (Fig. 3c).

Subsequently, we input each axial slice of the 3D RI images into our trained network (Fig. 3a). We cropped each wide-field RI image from every axial position into 1024 × 1024-pixel patches with 50% overlap. These patches were then fed into the trained neural network, which generated corresponding H&E images for each patch. The predicted virtual H&E patches were then stitched back together to reconstruct the original wide-field image (Fig. 3d). By repeating this prediction process for all axial positions, we successfully generated 3D virtual H&E images from the label-free, 10 μm-thick colon cancer tissue slide. The predicted H&E images effectively represented anatomical structures such as glands and lumens (Fig. 3e). Furthermore, the necrotic regions within glands were also depicted, as indicated by cyan arrows.

To delve further into the three-dimensional intricacies of colon cancer, we prepared an additional tissue slide from a separate patient, this time with a thickness of 20 μm—fivefold that of traditional slides. Utilizing the identical approach applied to the previously mentioned 10 μm-thick slide, we successfully captured 3D RI images and subsequently generated detailed 3D virtual H&E images (Fig. 3a). Similarly, the wide-field RI image effectively depicted the anatomical structures of glands and lumens (Fig. 4a). Detailed examination of selected lumens and glands in 3D clearly identified vacant regions of lumens, along with their surrounding glandular structures (Fig. 4b). Notably, as the sample thickness increased to 20 μm, the warping of lumens and glands along the axial axis became more evident, as indicated with yellow arrows. This dynamic presentation provides compelling evidence of the 3D reconstruction of anatomical structures of label-free 20 μm-thick colon cancer tissue slides.

Following the methodology used for 10 μm-thick tissue slide, label-free RI images of 20 μm-thick colon cancer tissue were input to the trained neural network to generate 3D virtual H&E images (Fig. 4c). The resulting virtual H&E images clearly depicted histological structures, with the warping of lumens along the axial axis as highlighted by yellow arrows (Fig. 4d). The outcomes underscore the capability of holotomography to accurately reconstruct the three-dimensional architecture of colon cancer tissues, handling tissue slides up to five times thicker than the standard. Furthermore, these findings emphasize the proficiency of our trained neural network in producing 3D virtual H&E images, showcasing its predictive strength in visualizing complex tissue structures without traditional staining methods.

### Subcellular examination of 3D virtual H&E images
Delving into the intricate three-dimensional outcomes produced by our trained neural network, we scrutinized the histological features in 3D. Cross-sectional analyzes unveiled the lumen and gland structures, with the lumens' void spaces consistently formed along the axial axis (Figs. 5a, b). These structures demonstrated noticeable structural

variations—such as widening, shrinking, or warping—along the z-axis. Moreover, the glandular formations surrounding these spaces were observed to change in width across different cross-sectional views. This variance underscores the 3D reconstruction of histological features within the virtual H&E images, affirming the network's capability to accurately replicate complex tissue architectures.

From the cross-sectional views, we observed distinct regions where the nuclei are either sparsely distributed or densely packed in 3D. For instance, in the uppermost cross-section of the 20 μm-thick sample result (Line 4 in Fig. 5b), nuclei are densely distributed across all axial sections, while in the middle cross-section of the same result (Line 5 in Fig. 5b), nuclei exhibit a relatively sparse distribution across most sections, except for a few initial axial sections.

Zooming in to the level of individual nuclei, we closely examined the 3D virtual H&E images alongside their corresponding input RI images. This comparison aimed to assess the fidelity with which the virtual staining process replicated the detailed structures and features of the nuclei within the tissue context (Figs. 5c, d). In our direct comparison, we observed circular structures of the nucleus in the input RI images, which is consistent with the predicted H&E images of blue circular morphologies. As we move along the axial axis, nuclei dynamically alter size and shape—contracting or expanding—indicating 3D construction of the nucleus. By comparing the virtual H&E images with the corresponding regions in chemical H&E images, we observed that while our approach effectively resolved both the shapes and axial positions of individual nuclei, the chemical H&E images were unable to achieve this, failing to distinguish single nuclei in the 20 μm thick tissue slide. Additionally, we examined the relative 3D distribution among nuclei by tracking the emergence and disappearance of some nuclei along the z-axis. For example, a nucleus indicated by a yellow arrow is located ~5.4 μm below a neighboring nucleus pointed by a cyan arrow (Fig. 5c). These results highlight the potential of 3D virtual H&E images to provide detailed insights into the 3D morphology and spatial distribution of cellular and subcellular structures, which can be easily overlooked in 2D analysis.

### Validation of the virtual H&E images with the chemically stained images
We further validated the capabilities of our method in the 3D histopathology of thick colon cancer tissue slides. To achieve this, we compared the 3D virtual H&E images to the chemically stained images of thick tissue slides (Fig. 6). After measuring label-free RI images of the two thick colon cancer slides, we chemically stained the same slide and obtained images using WSS. However, since the WSS images were only accessible in 2D, we needed to devise a strategy to generate representative 2D images from 3D virtual H&E images for a fair comparison. For this purpose, we utilized a single focal plane as one representative (Figs. 6a, d) and applied minimum intensity projection along the axial axis for another representative 2D image (Figs. 6b, e).

The comparison between virtual and chemical H&E images for the 10 μm-thick tissue slide highlighted the reliability and fidelity of the

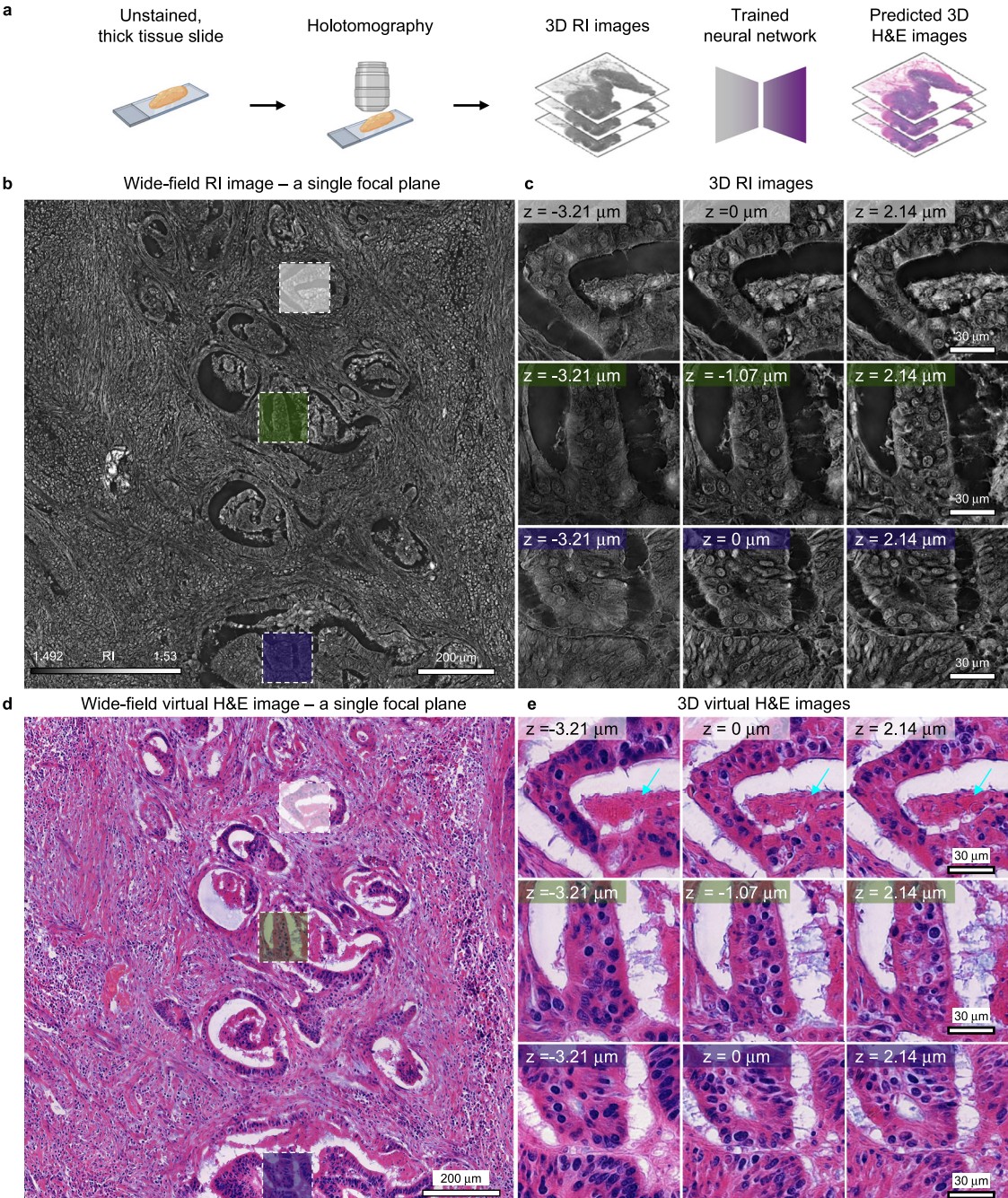

**Fig. 3 | Virtual 3D H&E staining of 10 μm-thick colon cancer tissue slide.**
**a** Workflow of generating 3D virtual H&E of 10 μm-thick colon cancer tissue slides
(**b**). A wide-field RI image obtained from the label-free 10 μm-thick colon cancer
tissue slides using holotomography. A single focal plane of the 3D RI image is
presented. **c** Detailed 3D RI images of glands and lumens obtained from 10 μm-

thick colon cancer tissue slides. **d** A wide-field virtual H&E image predicted from (**b**).
A single focal plane of the 3D virtual H&E image is presented. **e** Detailed images of
3D virtual H&E images. Cyan arrows indicate the necrotic structures. Figure 3a was
created with BioRender.com.

proposed virtual staining framework (Fig. 6a–c). The analysis of the
wide-field images exhibited the accurate reproduction of anatomical
features by the trained network. The anatomical structures including
glands and stroma are visible in the virtual H&E stained images. To
provide a detailed examination, we zoomed the glandular structures,
revealing comparable nucleus distributions to the chemically stained
slides (Fig. 6a–c (i, ii)).

Comparing virtual and chemical H&E for the 20 μm-thick tissue
slides demonstrated the robustness of the virtual staining framework,
even for sample thickness where the chemical staining becomes

abnormal (Fig. 6d–f). The colorimetric characteristics of virtual H&E
remain consistent with minimal deviation, while chemical staining
resulted in a distinctly different color distribution. Nevertheless, we
could compare the anatomical features between virtual and chemical
H&E images. The structures of glands and stroma were successfully
reproduced by the trained network. For a detailed investigation of
glandular structures, we zoomed into the selected glands, clearly
illustrating the consistency of lumens and surrounding glandular
structures (Fig. 6d–f (i, ii)). All zoomed-in regions are confirmed by the
experienced pathologist (SJS).

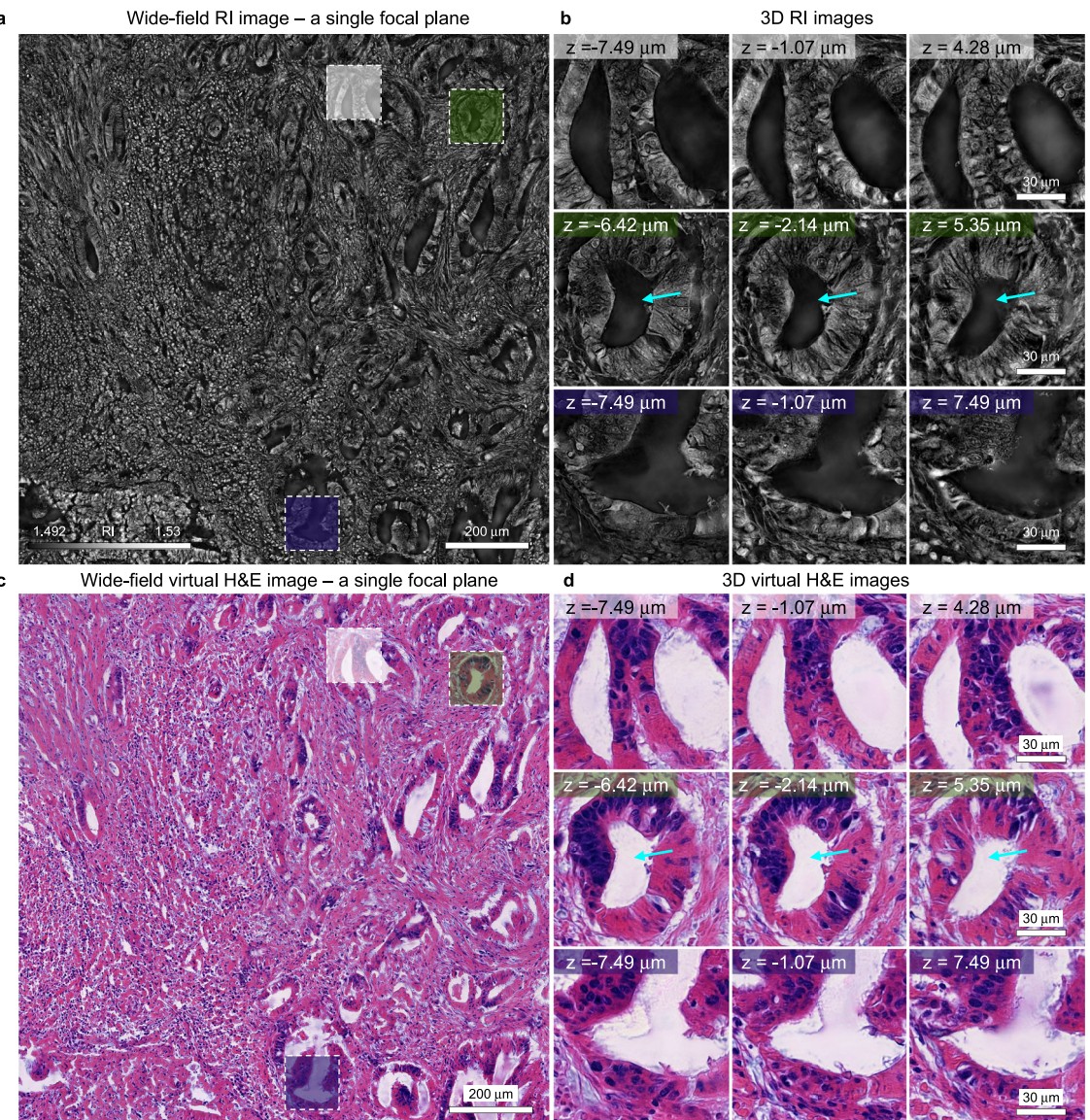

**Fig. 4 | Virtual 3D H&E staining of 20 µm-thick colon cancer tissue slide. a** A wide-field RI image obtained from the label-free 20 µm-thick colon cancer tissue slide using holotomography. A single focal plane of the 3D RI image is presented. **b** Detailed 3D RI images of glands and lumens obtained from the 20 µm-thick colon cancer tissue slide. **c** A wide-field virtual H&E image predicted from (**a**). A single focal plane of the 3D virtual H&E image is presented. **d** Detailed images of 3D virtual H&E images. Cyan arrows indicate lumens.

## 3D subcellular and microanatomical structures across the whole slide in colon cancer

To further investigate the volumetric structures of thick tissue, our framework was applied to a 50 µm-thick tissue slide. This label-free colon cancer tissue slide, obtained from a different patient, was prepared to a thickness of 50 µm (Fig. 7a). Through holotomography, we captured a whole slide 3D RI image, which were subsequently cropped and input into the same trained neural network to produce 3D virtual H&E images, with non-sample regions masked out. The results, encompassing a total volume of 15 mm × 15 mm × 0.05 mm, were compared with a WSS image of the same sample (Fig. 7b–d). Notably, for this experiment, we utilized a different holotomography system, situated at another institute compared to the equipment used in previous sections, to ensure the repeatability and robustness of our RI measurements.

Chemical H&E staining of thick tissue sections (50 µm) often exhibits color artifacts—such as overly bluish or reddish regions and

inconsistent tones—due to non-uniform staining (Fig. 7d, g). These artifacts highlight the limitations of conventional histology for thicker samples. In contrast, our virtual H&E approach produces more consistent color distributions (Figs. 7c, 7f) and offers enhanced analytical capabilities, including the ability to visualize the axial distribution of overlapping nuclei—features not readily discernible in standard chemical H&E images.

To closely examine the 3D lumen structure, we focused on detailed images of a lumen and surrounding gland within a region of interest, and compared it with the chemical H&E images (Fig. 7h–j). Through volumetric representation and multiple axial images, we elucidated the structural alterations of lumens as axial position changed. These alterations became more pronounced as the sample thickness increased compared to previous results. Furthermore, within the same region of interest, we identified the 3D distribution of nuclei and microanatomical structures by segmenting nuclei and the lumen (Fig. 7k). Leveraging the capabilities of Hover-Net[38], we automated the

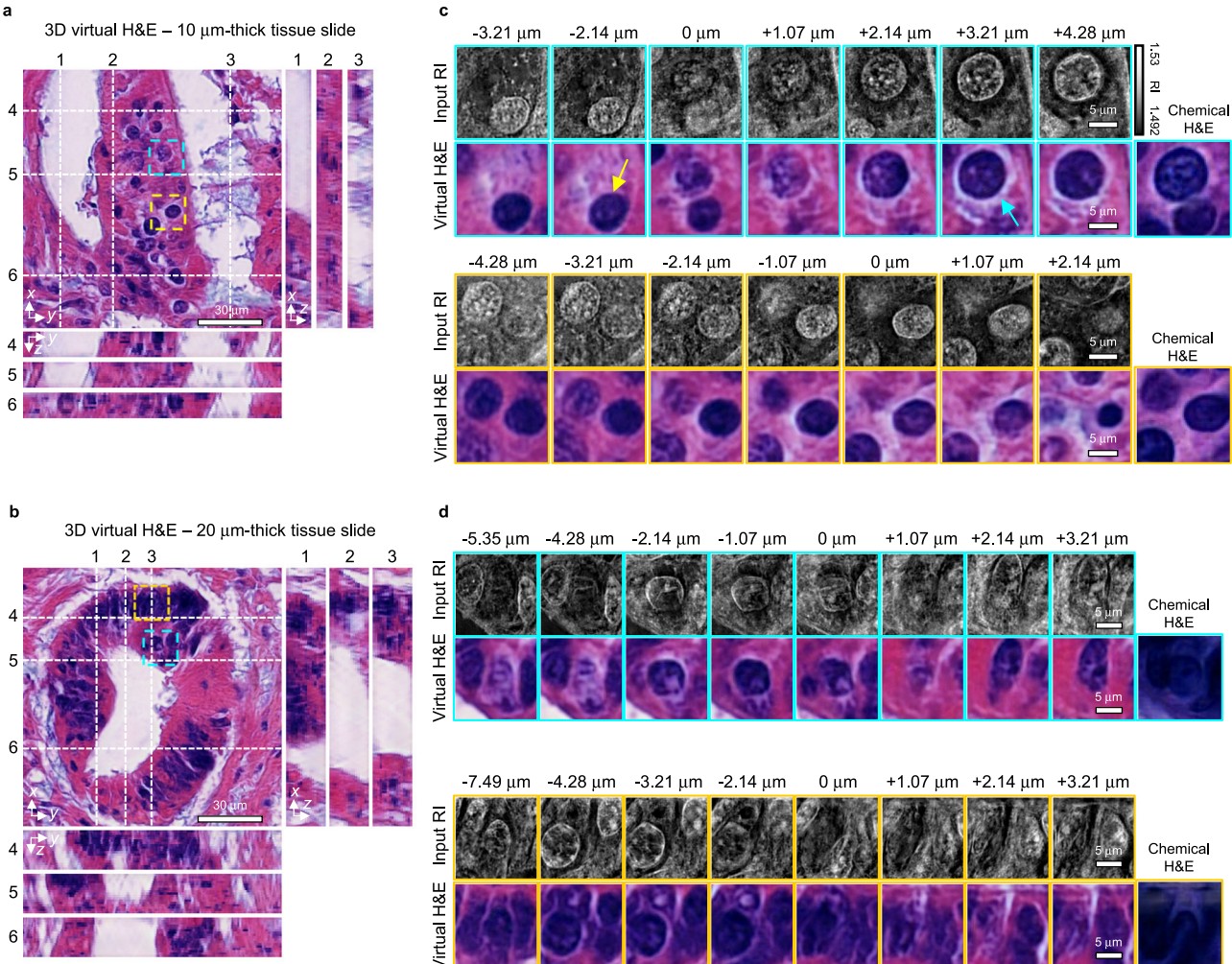

**Fig. 5 | 3D subcellular examination of colon cancer tissue slide a, b.** The *x–y*, *y–z*, and *x–z* cross sections of 3D virtual H&E images of the 10 (**a**) and 20 µm-thick colon cancer tissue slide (**b**). **c**, **d** Detailed images to visualize individual nuclei in input RI images and corresponding virtual H&E images from the 10 (**c**) and 20 µm-thick tissue slide (**d**) and their corresponding chemical H&E images. Yellow and cyan arrows indicate the individual nuclei at different axial positions.

segmentation of nuclei while labeling the lumens manually. With the resulting binary masks, we rendered the 3D microanatomical structures of lumens and the surrounding nuclei distribution (Fig. 7k).

To investigate deeper into subcellular structures, we extracted a cube encompassing a single nucleus, guided by the corresponding binary masks (Fig. 7l, m). This extraction not only unveiled the 3D structure of the nucleus but also provided quantitative morphological parameters such as volume and surface area. Additionally, we computed the average numbers, areas, and eccentricities of nuclei and traced their trends along the axial axis. Intriguingly, our analysis revealed a subtle increase in nuclear eccentricity, while their area and the number remained relatively consistent along the axial axis, with some observable fluctuations (Fig. 7n). Similarly, we quantified the perimeter, area, and major axis length of the lumen at each image slice. By tracking their variations along the axial axis, we unveiled a trend of contracting lumens as axial position increased, supported by a consistent decrease in these parameters. Specifically, we observed approximately a 40% reduction in the lumen area between the uppermost and lowermost axial sections (Fig. 7o). Unlike the quantitative analyzes of volumetric lumen structures enabled by 3D virtual H&E images, chemical H&E staining could not provide axial information or volumetric parameters of the lumens. Furthermore, due to distorted staining results, chemical H&E images were unable to resolve the nuclei distribution around the lumen (Fig. 7j).

These results not only provide our understanding of 3D colon cancer structure but also effectively highlight the scalability of 3D virtual staining, enabling a multitude of downstream quantitative analyzes on the 3D distribution of nuclei and microanatomical structures.

## Validation of virtual 3D H&E staining across sample origins and institutional settings

To ensure the scalability and repeatability of our virtual staining framework, validating its application across varied institutional settings and sample origins is essential. Our approach was tested in different countries, using gastric cancer slides imaged with holotomography at an institute distinct from those mentioned previously. The methodology for creating the training dataset mirrored that of the colon cancer studies, with variations only in the slide numbers and the types of WSS utilized (refer to Methods for details). This process yielded a dataset comprising 9231 patches, divided into 7002 for training and 2229 for validation. To quantitatively assess the network's performance, SSIM values were computed for each selected window (see Supplementary Fig. 3). Furthermore, to ensure methodological consistency, we maintained the same neural network architecture and hyperparameters as those used in the colon cancer study, reinforcing the consistency and robustness of our approach.

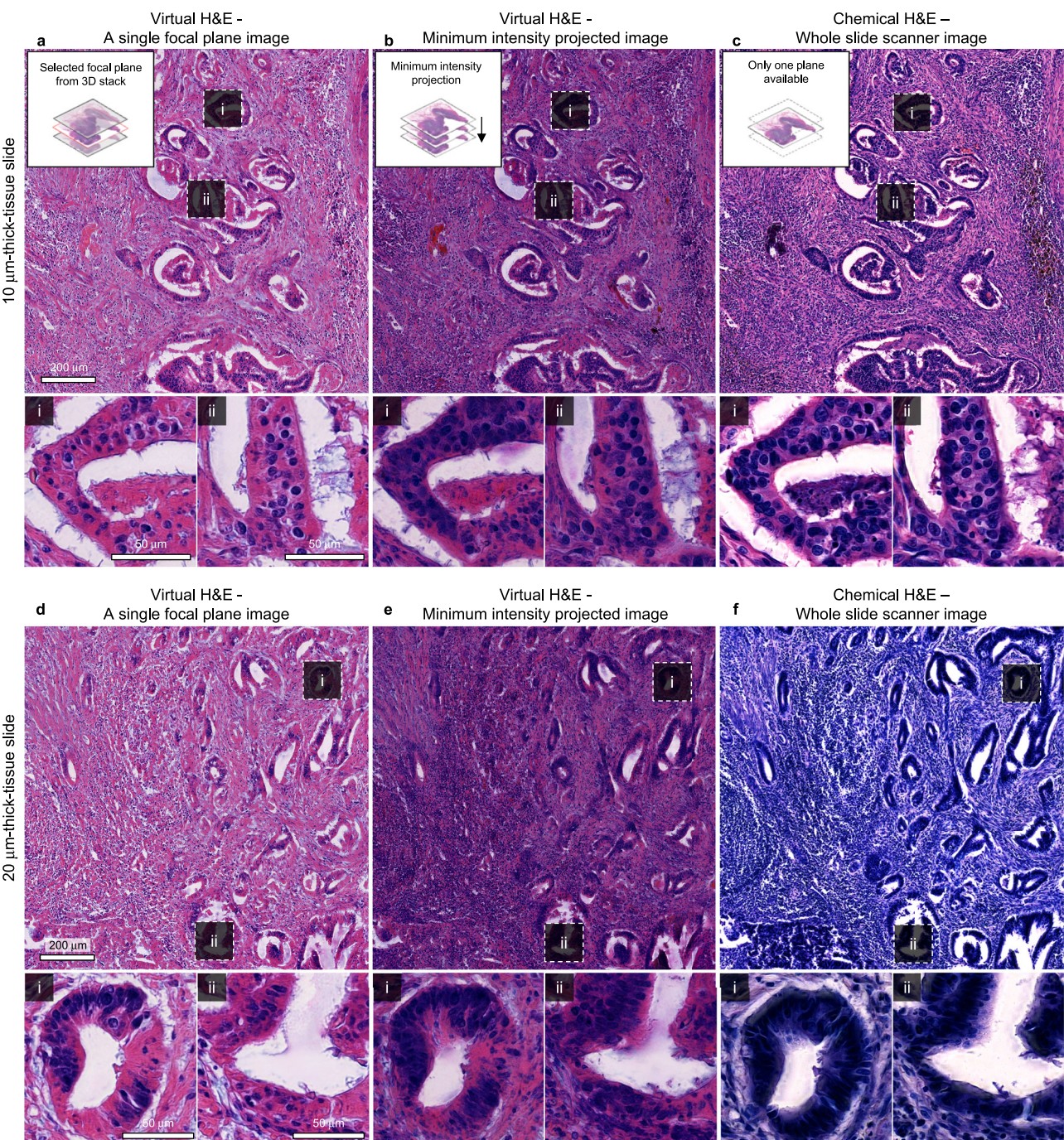

**Fig. 6 | Validations using the standard histopathology procedures. a, d** A single focal plane of 3D virtual H&E images predicted from label-free 10 (**a**) and 20 (**d**) μm-thick colon cancer tissue slide. **b, e,** Minimum intensity projection of 3D virtual H&E images predicted from label-free 10 (**b**) and 20 μm-thick colon cancer tissue slide (**e**). **c, f** Same 10 (**c**) and 20 μm-thick tissue slides (**f**) were stained with H&E and imaged using WSS. Detailed images of selected glandular structures are presented (i, ii). Data shown are from a single experiment.

Following training, we prepared a 20 μm-thick gastric tissue slide without any staining. Then, we acquired 3D RI images spanning ~1.92 × 1.22 mm regions using holotomography (Fig. 8a). These wide-field RI images clearly depicted anatomical structures such as smooth muscle and blood vessels, with vascular structures displaying individual cells within circular surrounding tissue structures pointed by a yellow arrow, alongside wavy textures in the smooth muscle as indicated with a cyan arrow (Fig. 8b).

Applying the same cropping and stitching procedures used for colon cancer data, we inputted the 3D RI images into our trained network to generate virtually stained 3D H&E images of the gastric cancer slides (Fig. 8c). Notably, the network effectively predicted most anatomical structures, including smooth muscle and vascular structures, as evidenced in wide-field images. Detailed examination revealed a successful depiction of vascular structures as red blood cells with surrounding circular connective tissue structures (a yellow arrow in Fig. 8d) and wavy textures of smooth muscle (cyan arrows in Fig. 8d). Leveraging the 3D nature of the data, we observed changes in the cross-sectional morphology of vascular structures along the axial axis, as well as alterations in the distribution of red blood cells

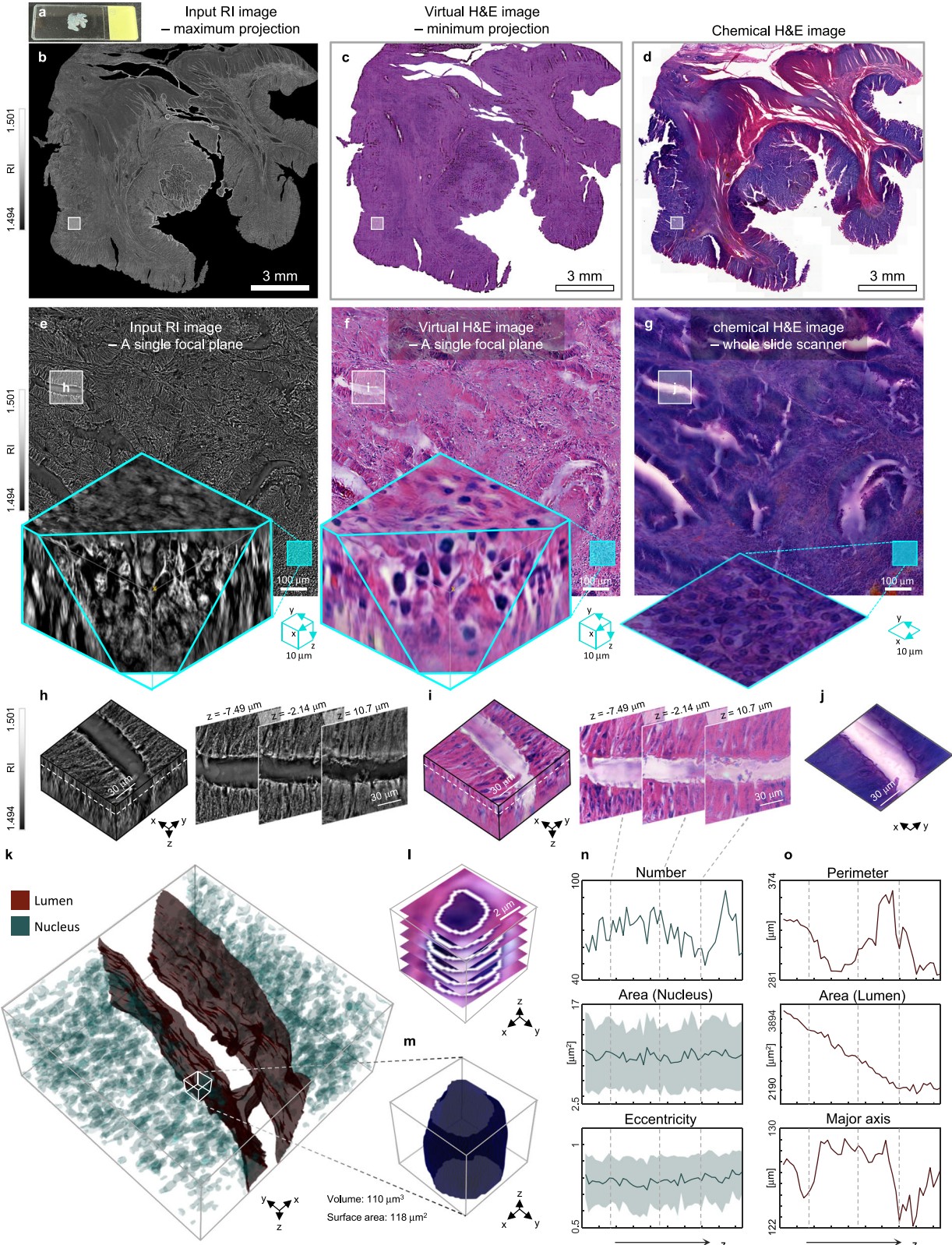

**Fig. 7 | 3D microanatomical rendering and quantitative analysis of a whole 50 μm-thick colon cancer tissue slide. a** A label-free colon cancer slide of 50 μm thickness used for imaging. **b** A maximum projected whole slide 3D RI image of (**a**). **c** A minimum projected whole slide 3D virtual H&E image of (**a**). **d** A WSS image of (**a**) after the staining. **e** Zoomed-in RI images of (**b**). **f** Zoomed-in virtual H&E images of (**c**). **g** Zoomed in WSS images of (**d**). **h** Detailed 3D RI images of the selected region. **i**, Detailed 3D virtual H&E images of the selected region. **j** Detailed WSS image of the selected region. **k** 3D rendering of the microanatomical structure of the selected region. **l** 3D binary masks of the single nucleus overlaid to the virtual H&E images. **m** 3D rendering of the nucleus using the binary masks in (**l**). **n** Tracking of average numbers, areas, and eccentricities of the nuclei along the axial axis. **o** Tracking of perimeter, area, and major axis length of the lumen along the axial axis.

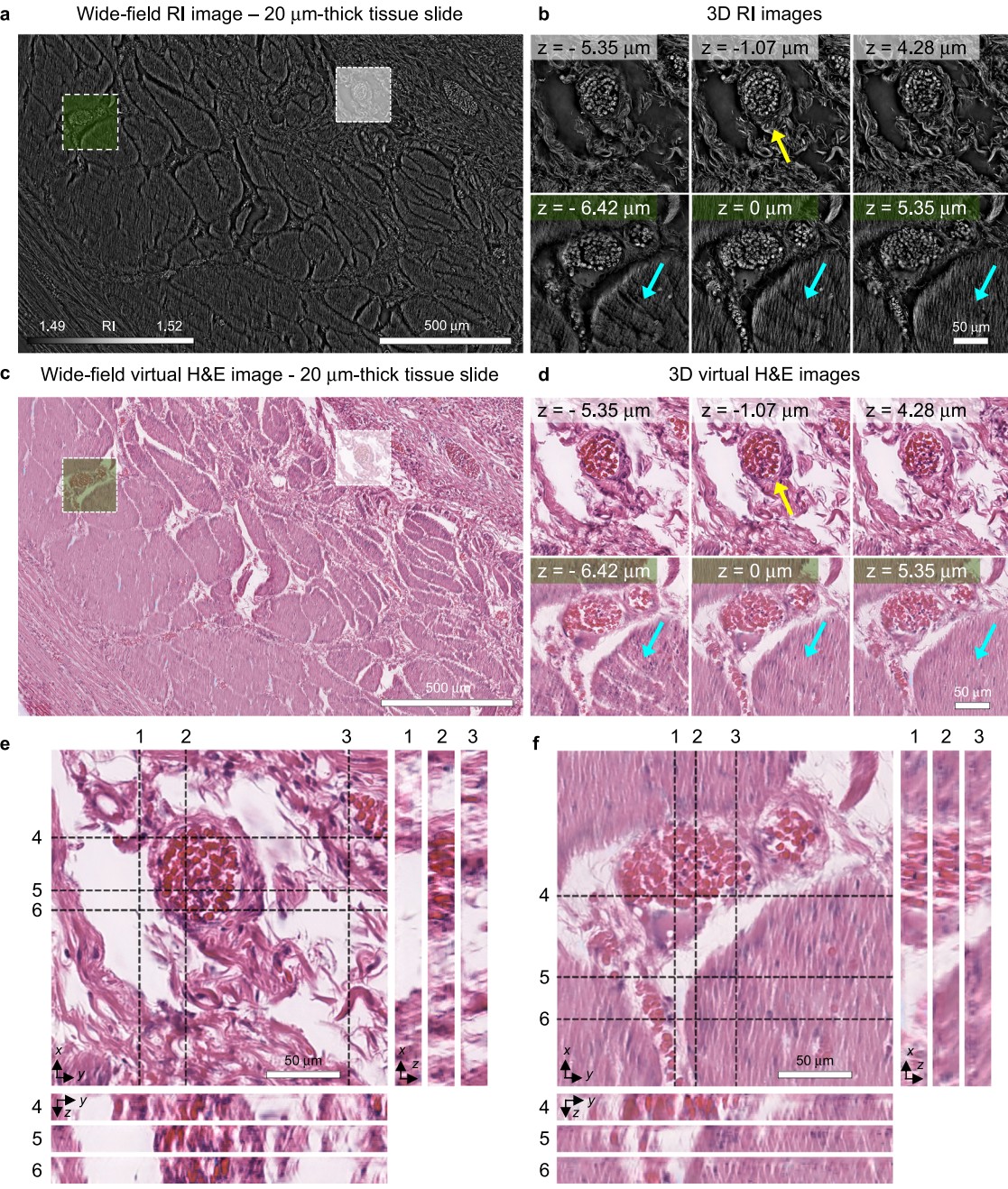

**Fig. 8 | Validations of the virtual staining across different organ types and institutional settings a.** A wide-field RI image obtained from the label-free 20 μm-thick gastric cancer tissue slide. A single section of the 3D RI image is presented. **b** Detailed 3D RI images of muscular (cyan arrows) and vascular (yellow arrows) structures obtained from the 20 μm-thick gastric cancer tissue slide. **c** A wide-field H&E image predicted from (**a**). A single section of the 3D H&E image is presented. **d** Detailed images of 3D H&E images predicted from (**a**). **e, f** The *x-y*, *y-z*, and *x-z* cross sections of predicted 3D H&E images.

within it. Moreover, variations in the wavy texture of smooth muscle along the axial axis were visualized as pointed by cyan arrows, highlighting the ability of virtual 3D H&E staining to reveal intricate 3D tissue textures.

To further elucidate structural features in 3D, we visualized cross-sections of detailed images depicting vascular and muscular structures (Fig. 8e, f). These cross-sectional views delineated continuous structural alterations along the axial axis in vessels and muscles, exhibiting 3D reconstruction. Additionally, we observed densely packed red blood cells throughout the whole axial sections (Lines 2 and 5 in Fig. 8e), as well as a widening tendency of smooth muscle as axial position increases (Lines 2 and 3 in Fig. 8f).

## Discussion

The developed virtual volumetric H&E staining framework adeptly overcomes two significant challenges: the labor-intensive nature of traditional sample staining and the limitations imposed by two-dimensional analysis. By training a neural network with RI images from a single, routinely prepared histopathological slide, we achieved the translation of these images into their H&E stained equivalents. This neural network, once applied to 3D RI images of label-free thick tissue, effectively produced 3D virtual H&E images. This method not only bypasses the need for physical staining, thereby reducing the requisite number of tissue slices, but also unveils detailed 3D representations of glandular, muscular, and vascular structures at the subcellular level

within thick cancer tissues. Additionally, this approach facilitates downstream analyzes, allowing for quantitative assessments and 3D visualizations of microanatomical structures and also for various multi-omics measurements including spatial transcriptomics, thus offering valuable insights into the three-dimensional architectural complexities of cancerous tissues.

The versatility of our virtual H&E staining method is particularly significant, as it is not confined to specific organ types or reliant on data training from any particular institution. This flexibility originates from a sample preparation, imaging, and network training process that is universally applicable, allowing for the method's scalability to tissue slides from a wide array of organs. Additionally, our approach is equally adaptable to cytology slides, which, despite their inherent 3D structures, have traditionally been examined in a 2D framework using bright-field microscopy. Our framework also proves its repeatability and reliability across various institutional settings, underscoring the robustness of both the holotomography technique and the neural network architecture employed. Such consistency further highlights the method's potential for broad application, promising valuable contributions to both research and clinical practices globally.

Building on this foundation, the proposed framework has significant potential to address a range of pathological challenges, such as detecting tumors in lymph nodes, identifying tumor-infiltrating lymphocytes, and assessing tumor margins. By offering comprehensive 3D views, this approach surmounts the limitations of conventional histology, which depends on 2D sections and may overlook critical features. Moreover, with further technical adaptations, the framework could be extended to immunohistochemistry or immunofluorescence, enabling more precise and specific assessments of pathological status in a 3D context.

Moreover, holotomography quantifies RI values, determining the protein concentration in specific regions. This empowers the proposed framework to conduct further quantitative analysis of subcellular structures. Specifically, segmentation or classification of cell types within the generated 3D H&E stained images can be performed using open-source, pre-trained resources[38], with results directly applicable to the RI images for calculating 3D morphological and biophysical parameters.

While our proposed virtual staining method has been successfully applied to tissues with a thickness of up to 50 μm, the applicable thickness can be pushed further through technical advances of holotomography. Currently, holotomography faces challenges related to image degradation as the sample becomes thicker. The degradation in image quality can be attributed to the multiple scattering of the light within the thick sample, which hinders the accurate reconstruction of RI values through holotomography. Nonetheless, several strategies are being employed to address this challenge, including enhancements to optical conditions, the implementation of reconstruction algorithms tailored for thicker samples[42], and the application of aberration correction algorithms[43]. Additionally, variations in standard sample preparation protocols—such as incomplete deparaffinization, selection of mounting media, and choice of fixatives—can influence the refractive index distribution and overall image quality. In particular, residual paraffin can introduce optical mismatches, while different fixatives may induce tissue shrinkage or dehydration, thereby affecting holotomography measurements. By adhering to standardized deparaffinization procedures, verifying the absence of residual paraffin, and carefully selecting or validating fixatives, we can mitigate these confounding factors. Optimizing these aspects of sample preparation and establishing robust quality-control measures will further improve the consistency and reliability of holotomography when integrated with virtual staining. With these refinements, we envision a synergistic advancement of both virtual staining and holotomography.

Beyond these challenges in image quality, scaling holotomography to whole-slide imaging introduces additional hurdles related to imaging speed and data storage requirements. This arises from its higher resolution, which is ~two to ten times greater than that of conventional approaches[2–5]. Recent advancements in imaging schemes[44,45], instrument design, and data processing methods[46] promise to enhance imaging speed and computational efficiency. By leveraging these developments, high-throughput holotomography can advance precise 3D tissue analyzes, thereby expanding its applications in both biomedical research and clinical practice.

With these capabilities and potential technical advancements, we believe that the proposed framework can effectively overcome current limitations in histopathology and immunology in unexplored dimentions[47,48], and broaden its utility across various histopathological studies[49,50], aiding in diagnosis and precision medicine.

## Methods

### Colon cancer slides preparation from Gangnam Severance Hospital

This study complied with all relevant ethical regulations and was approved by the Institutional Review Board (IRB) of Gangnam Severance Hospital (IRB No. 3-2022-0083) and Seoul National University Boramae Medical Center (IRB No. 30-2018-32). Written informed consent was waived by the IRB due to the retrospective nature of the study and the use of residual samples without any clinical data collection. To create the training datasets, we utilized a 4 μm-thick colon cancer tissue slide prepared as tumor microarrays. The slide was newly diagnosed and reviewed by a pathologist (SJS) at Gangnam Severance Hospital in January 2023. Among 26 cases, a colon cancer tissue slide with grade two was randomly selected. From the chosen case, a formalin-fixed paraffin-embedded tissue block was obtained. The tissue block was sliced to a 4 μm thickness using a microtome. The slice was carefully placed on a slide glass and deparaffinized using xylene. Subsequently, the slide was then stained with H&E, mounted with a mounting medium (Consul-Mount, Epredia), which has an RI value of 1.495 measured by a refractometer (R-5000, Atago), and finally covered with coverslips. Annotations of glands and stroma in the H&E stained images were carried out by an experienced pathologist (SJS).

For the label-free thick tissue slides used in the testing phase, we randomly selected two cases of colon cancer. Formalin-fixed paraffin-embedded blocks from the selected cases were sliced into 10, 20, and 50 μm-thick sections using a microtome, respectively. The subsequent procedures were consistent with those of training datasets, except that we did not stain the slides and mounted the slides immediately after the deparaffinization.

### Gastric cancer slides preparation from mayo clinic

For the training datasets, we obtained four gastric cancer tissue slides, each with a 5 μm thickness and H&E stain. The slide was newly diagnosed and reviewed by pathologists at the Seoul National University Boramae Medical Center in January 2006. The slide preparation procedures were consistent with those conducted at Gangnam Severance Hospital.

For testing purposes, a label-free, 20 μm-thick tissue slide was randomly selected from the gastric cancer cases. The slide was prepared using the same procedures as those performed at Gangnam Severance Hospital.

### Image acquisition

To measure the RI distributions of tissue slides, we employed a low-coherence holotomography setup (HT-X1 and X1-Plus, Tomocube). This system measures transmitted intensity images of a sample using four optimized Köhler illumination patterns[31], from which the 3D RI distribution was reconstructed by deconvolution of the intensity images with theoretically estimated point spread functions. In the experimental setup, a light-emitting diode (LED) with a center wavelength of 449 nm within a digital micromirror device (DMD) module

(DLP4500, Texas Instrument) was used as the illumination source. The blue wavelength was chosen to avoid overlap with the peak absorption spectrum of H&E staining. The illumination intensity was controlled using a DMD located in the Fourier plane, which was then projected onto the sample using a condenser lens ($f$ = 180 mm, numerical aperture (NA) = 0.75). The diffracted light from the specimen was collected using an objective lens (numerical aperture (NA) = 0.95) and measured using a CMOS camera (FS-U3-28S5, FLIR) located in the image plane. The RI images offer a lateral and axial resolution of 156 nm and 1.07 μm, respectively. In addition to RI images, this system offers single-channel BF (scBF) images, which are single-intensity images taken under uniform circular pattern illumination. Wide-field RI images were obtained by stitching the tiled RI images of each field of view with the size of $227 \times 227$ μm. For the stitching, the raw intensity images were captured with an overlap of 23 μm for both RI and scBF images. To obtain the ground truth images, the H&E stained slides were scanned using the whole slide scanner manufactured by 3D HISTECH for colon cancer slides and by Leica for gastric cancer slides.

## All-in-focus

We applied an all-in-focus algorithm only to the RI images obtained from a slide used for network training[32]. This algorithm ensures that the single focal plane includes most of the cellular structures. Specifically, we used a vignette size of 100 pixels with a step size of 10 pixels. Normalized variance was calculated in each window for all axial sections, and the window from the section with the maximum normalized variance value was selected as the best focal plane. Repeating the procedures for the whole 3D RI images resulted in a single all-in-focused RI image.

## Registration

Image registration between the RI and WSS images involved three steps. First, we manually cropped the regions from the WSS images where they were imaged using holotomography. Subsequently, we applied a spatial transform network[33] to register the manually cropped WSS images with the scBF images acquired using holotomography. This step resulted in paired images of scBF and WSS at a wide-field scale. Then, we cropped the paired datasets into $1024 \times 1024$-pixel patches and repeated the registration using a spatial transform network between these patches to ensure precise alignment at each patch scale. Note that aligning the WSS images with scBF images directly leads to the alignment with RI images, as scBF and RI images are captured at the same position.

All the registration steps, except for the manual cropping, were conducted using spatial transform networks[33]. Specifically, EfficientNet[51] was trained to discover the optimal affine transform matrix for the input WSS images, which will be used to transform the WSS images to be aligned with the target scBF images. The training was achieved by minimizing the loss defined using Pearson's correlation coefficient (PCC):

$$\mathcal{L}_{pcc}(x,y) = 1 - \frac{\mathbb{E}\left[(x - \mu_x)(y - \mu_y)\right]}{\sigma_x \sigma_y} \quad (1)$$

where $x$ and $y$ refer to the input WSS and target scBF images, respectively. $\mu$ and $\sigma$ denote the mean and the standard deviation of each image, respectively. Once the training was completed, we employed the trained network to predict the optimized affine transform matrix, which was then applied to register the WSS images.

## Training details

A conditional GAN was used to train the network for generating 3D H&E images, primarily based on a previous study[41]. In this framework, the generator's objective is to transfer the RI images into the feasible H&E stained BF images by minimizing the $\mathcal{L}_{pcc}$ between the images generated by the network and their corresponding ground truth images. Simultaneously, the discriminator aims to distinguish the images from the network and genuine images when provided with the corresponding input images. The final objective of this framework becomes

$$G^* = \arg \min_G \max_D \mathcal{L}_{cGAN}(G,D) + \lambda \mathcal{L}_{pcc}(G(x),y) \quad (2)$$

where $\mathcal{L}_{cGAN}$ is defined as:

$$\mathcal{L}_{cGAN}(G,D) = \mathbb{E}[og D(x,y)] + \mathbb{E}[\log(1 - D(x,G(x))] \quad (3)$$

where $G(\cdot)$ and $D(\cdot)$ refer to the generator and discriminator network operators, respectively. $x$ denotes an input image and $y$ indicates a ground truth image.

For the architecture of the generator, we used SCNAS[34], an optimized network specifically tailored for 3D medical image segmentation (Supplementary Fig. 2a). SCNAS employs a stochastic sampling algorithm within a gradient-based bi-level optimization framework to simultaneously search for the optimal network parameters at multiple levels using generic 3D medical imaging datasets. This search resulted in the discovery of a U-Net-like encoder-decoder structure with skip connections. In each micro-level architecture, we added a squeeze excitation block[52]. The key network parameters were as follows: activation function, leaky ReLU; normalization function, instance normalization; the size of the initial feature map, 12; the number of layers, 8; feature map multiplier, 2.

The discriminator consists of five convolutional layers with a kernel size of 4 and a stride of 2 (Supplementary Fig. 2b). The first four layers use leaky ReLU with a slope of 0.2, while the last layer employs ReLU as an activation function. Following the last layer, the result is activated by a Sigmoid function. The first and last layers use batch normalization, while the rest of the layers use instance normalization.

Both the generator and discriminator were trained using an adaptive moment estimation optimizer (ADAM) to update the learnable parameters[53]. Image augmentation techniques, such as adding blur, adjusting brightness, and introducing Gaussian noise, were randomly added to the training patches. A learning rate of $1 \times 10^{-4}$ was maintained throughout the training process. Training was conducted for 40 epochs with a batch size of 1, and early stopping was employed to prevent overfitting. The network was implemented using Python version 3.8.0 and PyTorch[54] version 1.13.1, and the procedures were executed on NVIDIA GeForce 4090 GPU.

The training results are compared with the ground truth images using SSIM, which is defined as below:

$$SSIM(x,y) = \frac{1}{3} \sum_{i=1,2,3} \frac{(2\mu_{x,i}\mu_{y,i} + C_1)(2\sigma_{xy,i} + C_2)}{\left(\mu_{x,i}^2 + \mu_{y,i}^2 + C_1\right)\left(\sigma_{x,i}^2 + \sigma_{y,i}^2 + C_2\right)} \quad (4)$$

where $x$ and $y$ are the network-generated output and corresponding ground truth image, respectively. $\mu$ and $\sigma$ refer to the mean and the standard deviation of each image, respectively, where an index $i$ refers to the RGB channels of each image. $\sigma_{xy,i}$ represents the covariance of both images calculated from $i$-th image channels. $C_1$ and $C_2$ are regularization constants, set as 6.5025 and 58.5225, respectively, by default. The SSIM values were calculated for the selected windows.

## Reporting summary

Further information on research design is available in the Nature Portfolio Reporting Summary linked to this article.

## Data availability

The training, testing, and exemplary data used in this study are available in the science data bank https://doi.org/10.57760/sciencedb.24217.

## Code availability

All neural network training, testing, and data analysis were performed using Python 3.8.0 and MATLAB R2021b. The code for developing the neural network is available in a GitHub repository https://github.com/BMOLKAIST/3D_virtual_HE_staining or https://doi.org/10.5281/zenodo.15030226[55] under the MIT License.

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

## Acknowledgements

The authors gratefully acknowledge the insightful comments provided by Seung-Mo Hong of Asan Medical Center. This work was supported by the National Research Foundation of Korea (RS-2024-00442348, 2022M3H4A1A02074314, RS-2023-00241278, RS-2024-00351903), an Institute of Information & Communications Technology Planning & Evaluation (IITP; 2021-0-00745) grant funded by the Korea government (MSIT), Korea Institute for Advancement of Technology (KIAT) through the International Cooperative R&D program (P0028463), and the Korean Fund for Regenerative Medicine (KFRM) grant funded by the Korea government (the Ministry of Science and ICT and the Ministry of Health & Welfare) (21A0101L1-12), Korea Health Technology R&D Project through the Korea Health Industry Development Institute (KHIDI), funded by the Ministry of Health & Welfare, Korea (HI21C0977, HR22C1605), National Cancer Institute (NCI) through grant 1R01CA276690 awarded to Tae Hyun Hwang (THH), grant 1R37CA265967 to THH, and grant U01CA294518 to THH. Additional support was provided by the Department of Defense (DOD) under grant CA190578 to THH. THH is also supported by the Eric and Wendy Schmidt Foundation's AI Innovation Award through the Mayo Clinic Foundation and the AACR Innovation and Discovery Grant.

## Author contributions

J.P. and S.-J.S. contributed equally to this work. J.P., G.K., H.C., D.R., D.A. and H.-S.M. developed the deep-learning pipeline, and J.P. implemented and optimized the pipeline. J.P., S.-J.S., J.C., I.B., M.K. and I.J., collected the data. S.-J.S., J.E.H., J.-Y.S., J.H.P., K.S.L and N.H.C. reviewed the pathology images. J.P., G.K. and Y.P. wrote the paper with critical inputs from all authors. T.H.H. and Y.P. supervised all aspects of the work.

## Competing interests

D.R., D.A., H.C., H.-S.M. and Y.K.P. have financial interests in Tomocube, a company that commercializes holotomography instruments. T.H.H. is a scientific co-founder of Kure.ai Therapeutics and its subsidiary, Kure.S. He holds no official roles in the companies and receives no salary or consulting fees. The companies did not influence the design, execution, or interpretation of this study. All other authors declare no competing interests.

## Additional information

[1]Department of Physics, Korea Advanced Institute of Science and Technology (KAIST), Daejeon, Republic of Korea. [2]KAIST Institute for Health Science and Technology, KAIST, Daejeon, Republic of Korea. [3]Department of Pathology, Gangnam Severance Hospital, Yonsei University College of Medicine, Seoul, Republic of Korea. [4]Tomocube Inc., Daejeon, Republic of Korea. [5]Department of Urology, Yonsei University College of Medicine, Seoul, Republic of

Korea. [6]Department of Artificial Intelligence and Informatics, Mayo Clinic, Jacksonville, FL, USA. [7]Department of Pathology, Kyung Hee University Hospital, Kyung Hee University College of Medicine, Seoul, Republic of Korea. [8]Department of Pathology, Seoul National University Boramae Medical Center, Seoul, Republic of Korea. [9]Department of Urology, Gangnam Severance Hospital, Yonsei University College of Medicine, Seoul, Republic of Korea. [10]Department of Pathology, Yonsei University College of Medicine, Seoul, Republic of Korea. [11]Florida Department of Health Cancer Chair, Mayo Clinic, Jacksonville, FL, USA. [12]Department of Immunology | Department of Cancer Biology, Mayo Clinic, Jacksonville, FL, USA. [13]Present address: Section of Surgical Sciences, Vanderbilt University Medical Center, Nashville, TN, USA. [14]These authors contributed equally: Juyeon Park, Su-Jin Shin.
✉e-mail: taehyun.hwang@vumc.org; yk.park@kaist.ac.kr

