## [Peer Review File · Nature Communications]

Revealing 3D microanatomical structures of unlabeled thick cancer tissues using holotomography and virtual H&E staining

Corresponding Author: Professor YongKeun Park

Version 0:

Reviewer comments:

Reviewer #1

(Remarks to the Author)

The authors report about a study on revealing 3D micro anatomical structures of unlabelled thick tissues from cancer patients using quantitative phase imaging (QPI)-based refractive index (RI) retrieval by holotomography in combination with virtual H&E staining via deep learning. After an introduction to the topic the method is firstly characterized experimentally by utilization of tissue slices. Then, experimental results from thicker tissue samples are presented.

In general, the presented research is motivated. The manuscript is organized, clearly written, and includes adequate references. The experimental investigations appear to be accurately performed. The results are original and plausible. The described approach represents an advance beyond the current state-of-the-art. The authors address an important topic in histopathology: Label free quantitative assessment of thicker tissues from cancer patients, which may be of interest for the areas of digital pathology and label free imaging as well as for the interdisciplinary field of quantitative phase microscopy. In summary, the content of the manuscript appears to be suitable for the journal Nature Communications.

However, the authors may consider revisions:

1. Sample preparation (main point): The authors should discuss the sample preparation steps and their possible impact on the virtual staining results and the consecutive image interpretation with more details:

a. Deparaffinization and tissue staining: In the methods section the authors write that they have applied deparaffinized tissue sections. Remaining paraffin (or different remaining paraffin amounts due to, e.g., variations of the paraffin removal protocol) can change the tissue refractive index which represents a central parameter in holotomography. Moreover, possible tissue fixation protocols may affect the tissue refractive index and/or cause RI variations. The authors should address these topics by a more extended discussion.

b. Mounting medium: The authors write that after the deparaffinization, the tissue slices are embedded for RI analysis with holotomography in a mounting medium which however seems to be not specified in descriptions of the manuscript. Interactions of the mounting medium with the tissue (depending on the chemical composition of the mounting medium) may/can change the tissue RI. Moreover, the RI of the mounting medium can influence the signal-to-noise ratio of the holotomographic QPI measurements via the refractive index difference to the tissue RI). The authors should specify the applied mounting medium/media and should add a more extended discussion concerning the possible influence of the applied mounting media on their method to the manuscript.

2. Standardization: With view to a potential application of the proposed method in diagnostics, the authors should discuss possible standardization demands, e.g., for the sample preparation and the training procedures of the neural network, to address, for example regulatory issues.

Other topics:

- Fig. 1e: For clarity, the authors should add denotations to the images in the figure.

(Remarks on code availability)

Reviewer #2

(Remarks to the Author)

In this work, Park et al introduce holotomography and virtual H&E staining as a means of creating 3D images of the microstructure of tissue samples. They posit that this technique has potential to be a valuable clinical tool for better understanding disease through 3D views, and that it is superior to existing 3D histology approaches because it is less time consuming and laborious. While the concept of reconstructing intact, H&E-stained volumes from holotomography-generated images of unstained samples is exciting, this work falls short of convincing this reviewer of the power of this approach. Particularly, I worry that many claims in this work oversell the authors results.

Major concerns:

The authors claim that this 3D technology represents a dramatic improvement over existing 2D clinical assessment. Yet the maximum volume of tissue they image is 1.2mm x 1.2mm x 50micron, or 0.072mm³. This is less volume than a typical "2D" whole slide image of (conservatively) 1.5cm x 1.5cm x 4micron, or 0.9mm³. The authors should state that their approach sacrifices the lateral size of conventional 2D histology for axial depth. The results section should focus on comparing the conclusions one would make when analyzing a single 2D whole slide image which has large lateral dimensions (1.5cm x 1.5cm) but no z axis depth, to their holotomography image, which has a much smaller lateral size (1.2mm x 1.2mm), and a modest z depth of up to 50micron. Does the addition of a third dimension improve assessment of clinical samples, even when the lateral size of the tissue is diminished? This question, which should be the central focus of the work, is not addressed.

In the introduction, the authors state "Current methods for achieving 3D histopathological analysis, however, entail labor-intensive preparation processes, including extensive sectioning, staining, or optical clearing." They claim their proposed method is superior because it is less labor intensive and does not require sectioning. Yet, the techniques they cite as inferior assess samples of mm³ or even cm³ volume. In this paper the maximum size of a sample assessed was 0.072mm³, hundreds of times smaller than these other 3D techniques. It is no surprise that acquisition of a small volume of tissue is less labor intensive than acquisition of a much larger volume. It is a misleading comparison to state that a method to assess micron³-sized samples is faster than a method to assess cm³-sized samples without stating the drastic difference in the volume of tissue obtained.

I found no mention of the pixel resolution (micron per pixel) of the holotomography images. If this technique is presented as a novel method for imaging 3D tissues, these specifications should be not only clearly stated but compared to conventional 2D histological imaging and current 3D imaging approaches.

I would like to see a ground-truth "chemical" H&E image compared to the virtual H&E image of the 50micron thick sample shown in Fig7b, as the morphology of these colon cancer glands is very different from that shown in earlier colon cancer samples. If there is a biological reason for this difference it should be explained. If this difference is due to poor performance of the imaging or virtual H&E staining, the authors must state that a sample of 50micron thickness may be beyond the current technical limitations of their approach.

Minor comments:

In Fig2E, the authors show in a violin plot that the SSIM ranged from roughly 0.75 – 0.82. In Figure 2C they show image examples of these SSIM values, choosing four examples with very high SSIM (0.83, 0.81, 0.81, and 0.8). It is misleading to show only top performing regions as the virtual staining algorithm does not perform at this quality across the entire sample. I suggest a mix of over-achieving, average, and underachieving regions of interest are shown.

(Remarks on code availability)

Reviewer #3

(Remarks to the Author)

The article presents an approach to acquire 3-D WSIs volumes by combining holotomography and H&E virtual staining supported by deep neural network dedicated to style/modality transfer. Overall, the method may be of great importance and I see its tremendous value, nevertheless, I have the following questions that should be answered and supported by additional ablation studies:

1) One of the main contributions of the article is related to the H&E virtual staining. Is there any particular reason why it was decided to use GAN-based approach to train the virtual staining network? What about direct encoder-decoder architectures, latent diffusion models or contrastive methods? According to the current state-of-the-art, these methods outperform GAN-based style/modality transfer in majority of scientific challenges. The article should report a comparison of these methods for the virtual staining.

- 2) How well does the method generalize to previously unseen data? What would happen if the virtual staining would be applied to different tissues?
- 3) The H&E staining is not directly standardized and the staining of the same tissue sample at different institutions may result in considerably different appearance. How does the proposed method solve the problem? Is there a mechanism to account for the variable concentration of hematoxylin and eosin for "really-stained" samples?
- 4) Is the holotomography free from slicing artifacts and distortions? How does the virtual staining method behave when it encounters an artifacts caused by the slicing procedure?
- 5) How do the authors ensure that the virtual staining method does not hallucinate non-existing structures?
- 6) The distribution of the SSIM is mostly between 0.75 and 0.82. Is it enough for the downstream tasks? How will the virtually stained 3-D volume work for AI-supported downstream tasks that were trained using the "traditional" 3-D reconstruction methods based on slicing and reconstruction based on image registration? Would it require to retrain or fine-tune already existing architectures to the virtually stained 3-D volumes?
- 7) Can the authors quantitatively estimate the benefits of the proposed method? How much does it reduce the costs and time associated with the 3-D reconstruction compared to the traditional methods?
- 8) How have the authors ensured that the registration used to create the ground-truth paired volumes is correct? The applied registration method seems like a relatively basic and error-prone method. What about using VALIS, RegWSI, or HistokatFusion?
- 9) How can the method be transferred to IHC virtual staining? Does it need to retrain the model for each stain separately? Are there other options? I suggest to introduce a brief description of that in the article.

(Remarks on code availability)

The code is very basic, however, it should allow to reproduce the results (by performing the inference on the exemplary cases). The model and exemplary data are released, however, the training cannot be reproduced without releasing the full training dataset.

Reviewer #4

(Remarks to the Author)

- integration of various growth patterns e.g., mucinous, signet ring cell? Network trained on one 4um thick colon cancer slide. How well is different differentiation patterns represented in virtual HE? integration of normal mucosa commonly found on whole slide sections? how well does the method represent necrosis or apoptotic regions, regions of very crowded nuclei?
- how does the method deal with samples prepared in different fixation methods from different institutions or how does age impact applicability of the technology?
- i agree one of the primary purposes of H&E is to facilitate visualisation of nuclei and cytoplasm but also nuance of sub-nuclear features such as chromatin intensity. how did these features compare between chemical and virtual?
- the authors suggest no significant difference between the number and size of nuclei between chemical and virtual however a trend is observed for average nucleus number ($p=0.0814$). This is likely insufficient for clinical application of such technology. There is a wider distribution of nuclei between patches for virtual compared to chemical - were there any specific features in those with larger discordance between chemical and virtual?
- minor comment - in fig 4d i would move the yellow arrow to point at the lumen - currently it sits on top of the epithelium.
- line 280 - "In our direct comparison, we observed circular structures of the nucleus in the input RI images, which is consistent with the predicted H&E images of blue circular morphologies" - these comparisons are not made between virtual method and ground truth of chemical staining - it would be nice to include the canonical chemical stain comparison
- line 299 - did the authors consider serial reconstruction of the HE for a more representative "tangible" comparison between virtual 3D and chemical 3D?
- Line 315 - comparing 20um thick HE with virtual staining - the authors comment on the chemical staining becoming abnormal at 20um, does this represent a fair comparison of the virtual to the ground truth? a significantly higher cellularity can be observed in fig 6c compared to 6a.
- what is the scalability of this method? From a clinical utility perspective - number of samples per run, time per scan?
- the authors discuss the potential for this technology to overcome current limitations in histopath and immunology to aid in diagnosis and precision medicine - it would be nice to add some sort of clinical comparison e.g., use case to elaborate on this. For example: How does the application of the 3D imaging improve detection of tumour in lymph nodes, if applied to a small clinical cohort what added value does this visualisation method bring for aiding staging? Or quantitation of tumour infiltration lymphocytes?

(Remarks on code availability)

Version 1:

Reviewer comments:

Reviewer #1

(Remarks to the Author)

In the revised manuscript, the authors have addressed my comments and questions.

(Remarks on code availability)

Reviewer #2

(Remarks to the Author)

The authors have addressed my major concerns, and I am pleased with the revised manuscript and the addition of a significantly larger specimen in their results. I recommend this work for publication.

(Remarks on code availability)

I have confirmed that the code files are present at the link provided by the authors, but I have not downloaded or tried to run the programs myself.

Reviewer #3

(Remarks to the Author)

Authors successfully addressed all my previous comments related to missing ablation studies, generalizability, artifacts, and potential benefits.

(Remarks on code availability)

The code should allow to reproduce the results. The model and exemplary data are released.

Reviewer #4

(Remarks to the Author)

I feel the authors have considered and addressed all the point raised in the initial review.

(Remarks on code availability)

REVIEWER COMMENTS

Reviewer #1 (Remarks to the Author): Expert in 3D imaging and tomography

The authors report about a study on revealing 3D micro anatomical structures of unlabelled thick tissues from cancer patients using quantitative phase imaging (QPI)-based refractive index (RI) retrieval by holotomography in combination with virtual H&E staining via deep learning. After an introduction to the topic the method is firstly characterized experimentally by utilization of tissue slices. Then, experimental results from thicker tissue samples are presented.

In general, the presented research is motivated. The manuscript is organized, clearly written, and includes adequate references. The experimental investigations appear to be accurately performed. The results are original and plausible. The described approach represents an advance beyond the current state-of-the-art. The authors address an important topic in histopathology: Label free quantitative assessment of thicker tissues from cancer patients, which may be of interest for the areas of digital pathology and label free imaging as well as for the interdisciplinary field of quantitative phase microscopy. In summary, the content of the manuscript appears to be suitable for the journal Nature Communications.

We thank the reviewer for their positive and encouraging feedback on our manuscript. We appreciate the recognition of our contribution to the label-free assessment of thicker tissues in histopathology and its potential impact.

However, the authors may consider revisions:

1. Sample preparation (main point): The authors should discuss the sample preparation steps and their possible impact on the virtual staining results and the consecutive image interpretation with more details:

[1]

a. Deparaffinization and tissue staining: In the methods section the authors write that they have applied deparaffinized tissue sections. Remaining paraffin (or different remaining paraffin amounts due to, e.g., variations of the paraffin removal protocol) can change the tissue refractive index which represents a central parameter in holotomography. Moreover, possible tissue fixation protocols may affect the tissue refractive index and/or cause RI variations. The authors should address these topics by a more extended discussion.

We thank the reviewer for highlighting the potential influence of residual paraffin and tissue fixation protocols on refractive index (RI) measurements, which are central to holotomography. Indeed, incomplete removal of paraffin may introduce RI mismatches between paraffin and the tissue,

potentially affecting both the measured morphology and quantitative RI values. To address this concern, we have (1) followed standardized deparaffinization procedures routinely employed in clinical pathology laboratories to mitigate variability and (2) verified the absence of residual paraffin through standard quality-control measures (e.g., inspecting tissue sections under bright-field microscopy after the final rehydration step). These steps are now detailed in the revised manuscript.

Moreover, we acknowledge that different fixation protocols—such as the use of methanol, ethanol, or 4% paraformaldehyde—can cause variations in tissue shrinkage and hydration status, ultimately influencing RI. While our primary goal in this study was to establish a generalizable framework, we have expanded our discussion in the revised manuscript to address how these fixatives could affect RI measurements. Specifically, we note that (1) most standard fixatives alter tissue density in predictable ways, and (2) our method tolerates small, systematic changes in RI without substantial loss of accuracy. Nonetheless, future investigations could explore whether additional model training or recalibration is warranted for tissues fixed under markedly different conditions.

[2]

b. Mounting medium: The authors write that after the deparaffinization, the tissue slices are embedded for RI analysis with holotomography in a mounting medium which however seems to be not specified in descriptions of the manuscript. Interactions of the mounting medium with the tissue (depending on the chemical composition of the mounting medium) may/can change the tissue RI. Moreover, the RI of the mounting medium can influence the signal-to-noise ratio of the holotomographic QPI measurements via the refractive index difference to the tissue RI). The authors should specify the applied mounting medium/media and should add a more extended discussion concerning the possible influence of the applied mounting media on their method to the manuscript.

We thank the reviewer for highlighting this important aspect of holotomography imaging. In the revised manuscript, we have specified the mounting medium (including its manufacturer and refractive index) and expanded our discussion on how it may influence both the measured RI values and the signal-to-noise ratio. In particular, we note that the mounting medium's refractive index, as well as any potential interactions with the tissue (e.g., partial tissue dehydration or swelling), could introduce subtle variations in the observed RI distribution. We have also included considerations for matching the mounting medium's refractive index to that of the tissue in order to minimize image artifacts. This additional discussion outlines both current best practices and potential strategies for optimizing or calibrating the mounting medium to ensure consistency and accuracy in holotomographic QPI measurements.

[3]

2. Standardization: With view to a potential application of the proposed method in diagnostics, the authors should discuss possible standardization demands, e.g., for the sample preparation and the training procedures of the neural network, to address, for example regulatory issues.

We thank the reviewer for emphasizing the importance of standardization for eventual diagnostic applications. As noted in our earlier responses, variations in sample preparation protocols—both within and across different institutions—can affect neural network performance. To address these challenges, we advocate for systematic validation of protocol parameters (e.g., fixation methods, deparaffinization steps, and mounting media) and the establishment of best-practice guidelines that promote reproducibility.

In developing our neural networks, we have made our training datasets, code, and current models publicly available and have documented the training pipeline in detail. We view these steps as integral to building a standardized framework, as they enable researchers and regulatory bodies to evaluate, reproduce, and refine our methods. Furthermore, as part of meeting regulatory requirements, it may be necessary to demonstrate consistent performance across multiple clinical sites using independent sample cohorts. We have expanded our discussion of these issues in the revised manuscript, highlighting how standardized protocols and transparent documentation can facilitate both broader clinical adoption and regulatory approval of our framework.

[4]

Other topics:

- Fig. 1e: For clarity, the authors should add denotations to the images in the figure.

We thank the reviewer for their recommendation. We have added denotations to Figure 1e in the revised manuscript.

Reviewer #2 (Remarks to the Author): Clinical expert in cancer digital pathology, 3D pathology, and AI

In this work, Park et al introduce holotomography and virtual H&E staining as a means of creating 3D images of the microstructure of tissue samples. They posit that this technique has potential to be a valuable clinical tool for better understanding disease through 3D views, and that it is superior to existing 3D histology approaches because it is less time consuming and laborious. While the concept of reconstructing intact, H&E-stained volumes from holotomography-generated images of unstained samples is exciting, this work falls short of convincing this reviewer of the power of this approach. Particularly, I worry that many claims in this work oversell the authors results.

We sincerely thank the reviewer for their thoughtful and constructive feedback on our manuscript. We appreciate the opportunity to refine and improve our work, particularly by expanding the experimental results and ensuring a balanced presentation of our findings and claims.

Major concerns:

[1]

The authors claim that this 3D technology represents a dramatic improvement over existing 2D clinical assessment. Yet the maximum volume of tissue they image is 1.2mm x 1.2mm x 50micron, or 0.072mm³. This is less volume than a typical “2D” whole slide image of (conservatively) 1.5cm x 1.5cm x 4micron, or 0.9mm³. The authors should state that their approach sacrifices the lateral size of conventional 2D histology for axial depth. The results section should focus on comparing the conclusions one would make when analyzing a single 2D whole slide image which has large lateral dimensions (1.5cm x 1.5cm) but no z axis depth, to their holotomography image, which has a much smaller lateral size (1.2mm x 1.2mm), and a modest z depth of up to 50micron. Does the addition of a third dimension improve assessment of clinical samples, even when the lateral size of the tissue is diminished? This question, which should be the central focus of the work, is not addressed.

We thank the reviewer for this insightful feedback. In the revised manuscript, we have significantly expanded our imaging scope to demonstrate that our approach need not sacrifice lateral coverage for axial depth. Specifically, we now present a representative large-scale dataset with a tissue volume of approximately 15 mm × 15 mm × 0.05 mm (for a total of 11.25 mm³)—156 times larger than the original 1.2 mm × 1.2 mm × 0.05 mm sample volume. By stitching multiple high-resolution tiles together, our holotomography-based framework offers volumetric microanatomical information, such as the 3D organization of nuclei and lumens, while covering an area comparable to a conventional whole-slide image (Figure R1).

Furthermore, to address the reviewer’s central question—whether the addition of a third dimension meaningfully enhances clinical assessment—we include a new discussion contrasting the conclusions drawn from a single 2D section with those enabled by 3D holotomography. Our results illustrate that the

third dimension can reveal morphological features (e.g., complex glandular structures) not readily apparent in a single 2D plane. While this benefits diagnostic interpretation, we also acknowledge the technical trade-offs. In particular, we discuss the increased acquisition time, data size, and computational overhead needed for 3D whole-slide imaging in holotomography. We propose strategies such as parallelizing acquisitions and optimizing compression algorithms to mitigate these challenges, which we believe will further expand the practicality of our 3D imaging approach in clinical settings.

By expanding coverage, providing volumetric insight, and outlining a clear path to addressing technical constraints, we hope to convey a balanced perspective on the benefits and trade-offs of 3D holotomography relative to conventional 2D histology.

Figure R1 | 3D microanatomical rendering and quantitative analysis of a whole 50 μm -thick colon cancer tissue slide a, A label-free colon cancer slide of 50 μm thickness used for imaging. b, A maximum projected whole slide 3D RI image of a. c, A minimum projected whole slide 3D virtual H&E image of a. d, A WSS image of a after the staining. e, Zoomed-in RI images of b. f, Zoomed-in virtual H&E images

of c. g, Zoomed in WSS images of d. h, Detailed 3D RI images of the selected region. i, Detailed 3D virtual H&E images of the selected region. j, Detailed WSS image of the selected region. k, 3D rendering of the microanatomical structure of the selected region. l, 3D binary masks of the single nucleus overlaid to the virtual H&E images. m, 3D rendering of the nucleus using the binary masks in l. n, Tracking of average numbers, areas, and eccentricities of the nuclei along the axial axis. o, Tracking of perimeter, area, and major axis length of the lumen along the axial axis.

[2]

In the introduction, the authors state “Current methods for achieving 3D histopathological analysis, however, entail labor-intensive preparation processes, including extensive sectioning, staining, or optical clearing.” They claim their proposed method is superior because it is less labor intensive and does not require sectioning. Yet, the techniques they cite as inferior assess samples of mm³ or even cm³ volume. In this paper the maximum size of a sample assessed was 0.072mm³, hundreds of times smaller than these other 3D techniques. It is no surprise that acquisition of a small volume of tissue is less labor intensive than acquisition of a much larger volume. It is a misleading comparison to state that a method to assess micron³-sized samples is faster than a method to assess cm³-sized samples without stating the drastic difference in the volume of tissue obtained.

We thank the reviewer for pointing out the ambiguity in our original statement. To clarify, our reference to “labor-intensive processes” in the introduction refers specifically to sample preparation—namely, the repeated sectioning, staining, and optical clearing required by many existing 3D histopathological methods. In contrast, our approach requires only a single tissue section without the need for staining or clearing, significantly reducing the complexity of the preparation phase. To avoid misinterpretation, we have added the term “sample preparation” in the revised manuscript.

We also acknowledge the reviewer’s valid observation regarding the difference in tissue volume between our method and other cited approaches. It is true that imaging smaller tissue volumes requires less time and resources overall. However, as we discuss in response to Reviewer #1 and have now expanded in the revised manuscript, once our framework is scaled to perform whole-slide imaging, acquisition times can extend significantly—potentially on the order of days. This increase results from the high resolution of holotomography (approximately 0.156 μm), which is two to ten times finer than that of the referenced methods (0.29–1.5 μm). While this finer resolution inevitably demands more time, it also yields substantially enhanced structural detail that can be pivotal for accurate 3D tissue analysis.

We have now clarified these points in the manuscript to ensure there is no confusion about how sample volume and resolution requirements each contribute to overall workflow complexity and imaging speed. We hope this addresses the reviewer’s concerns regarding volume comparisons and the labor-intensity of our approach.

[3]

I found no mention of the pixel resolution (micron per pixel) of the holotomography images. If this technique is presented as a novel method for imaging 3D tissues, these specifications should be not only clearly stated but compared to conventional 2D histological imaging and current 3D imaging approaches.¹

We thank the reviewer for highlighting the need to clearly specify and compare our pixel resolution. In the revised manuscript, we have emphasized that the lateral resolution, or the diffraction limited optical resolution, of our holotomography system is approximately 156 nm (0.156 μm) per pixel. To place this in context, we now explicitly compare it to the resolutions reported in conventional 2D histology and other 3D imaging approaches:

- Conventional 2D histology imaging and Reference 1¹: 0.5 $\mu\text{m}/\text{pixel}$
- Reference 2²: 0.29 $\mu\text{m}/\text{pixel}$
- Reference 3³: ~ 0.5 $\mu\text{m}/\text{pixel}$ (average)
- Reference 4⁴: 1.5 $\mu\text{m}/\text{pixel}$

This demonstrates that our method achieves a lateral resolution roughly two to ten times finer than the referenced techniques. While this increased resolution can result in slower imaging speeds, it enables detailed visualization of morphological and subcellular structures, as shown in Figure 5. We have incorporated these details into our discussion of imaging speed and scalability to address the reviewer's concern.

[4]

I would like to see a ground-truth "chemical" H&E image compared to the virtual H&E image of the 50micron thick sample shown in Fig7b, as the morphology of these colon cancer glands is very different from that shown in earlier colon cancer samples. If there is a biological reason for this difference it should be explained. If this difference is due to poor performance of the imaging or virtual H&E staining, the authors must state that a sample of 50micron thickness may be beyond the current technical limitations of their approach.

We thank the reviewer for this insightful comment and the opportunity to clarify our results. To directly compare the morphology of our 50 μm thick colon cancer sample, we performed chemical H&E staining on the exact same tissue section used for holotomography (Figure R1). Notably, the chemical H&E image revealed significant artifacts—such as overly bluish or reddish areas and patchy staining—arising from the difficulty in uniformly staining such a thick sample (Figures R1b–d). These artifacts underscore the technical challenges inherent to conventional histology on thick sections.

In contrast, our virtual H&E approach avoids these staining artifacts while providing additional volumetric information, including the axial distribution of overlapping nuclei (Figures R1e–g). The distinctive gland morphology shown in Figure R1 arises from sampling a cross-section that includes the gland's internal lumen, rather than representing a shallow or tangential slice, as is more common in thinner sections. This structural difference has been verified by an experienced pathologist (SJS), who confirmed that it reflects true biological variation rather than poor imaging or staining performance.

However, we acknowledge that imaging thicker sections can pose technical challenges. If performance issues were observed—such as reduced resolution or inaccurate virtual staining—we would note that 50 μm might exceed current technical constraints. In this case, our side-by-side comparison of chemical and virtual H&E images supports that the observed morphological differences are indeed biological and not attributable to limitations in the holotomography or the virtual staining method. A discussion of these points has been added to the Results section to address the reviewer's comments.

Minor comments:

[5] In Fig2E, the authors show in a violin plot that the SSIM ranged from roughly 0.75 – 0.82. In Figure 2C they show image examples of these SSIM values, choosing four examples with very high SSIM (0.83, 0.81, 0.81, and 0.8). It is misleading to show only top performing regions as the virtual staining algorithm does not perform at this quality across the entire sample. I suggest a mix of over-achieving, average, and underachieving regions of interest are shown.

We thank the reviewer for suggesting a more balanced presentation of our results. In the revised manuscript (Figure 2E), we now include examples from different performance ranges to provide a more comprehensive view of the virtual staining algorithm's variability. Specifically, we have retained two patches with SSIM values greater than 0.80 and added two additional patches at lower performance levels (SSIM values of 0.77 and 0.78). By incorporating these underperforming examples, we aim to illustrate both the high and modest ends of our algorithm's output, thereby offering a clearer depiction of its overall performance.

Reviewer #3 (Remarks to the Author): Expert in deep learning, virtual image reconstruction, and 3D vision

The article presents an approach to acquire 3-D WSIs volumes by combining holotomography and H&E virtual staining supported by deep neural network dedicated to style/modality transfer. Overall, the method may be of great importance and I see its tremendous value, nevertheless, I have the following questions that should be answered and supported by additional ablation studies:

We sincerely thank the reviewer for recognizing the potential value of our approach. We appreciate the opportunity to address the technical details to further enhance the scientific rigor and clarity of our work.

[1]

One of the main contributions of the article is related to the H&E virtual staining. Is there any particular reason why it was decided to use GAN-based approach to train the virtual staining network? What about direct encoder-decoder architectures, latent diffusion models or contrastive methods? According to the current state-of-the-art, these methods outperform GAN-based style/modality transfer in majority of scientific challenges. The article should report a comparison of these methods for the virtual staining.

We appreciate the reviewer's interest in why we chose a GAN-based approach for virtual H&E staining. While we initially adopted a GAN framework due to its strong performance in style transfer tasks and its proven ability to preserve fine structural details, we have now conducted additional comparative experiments with U-Net, Pix2Pix, and conditional diffusion-based architectures (Figure R2). These tests revealed that all methods perform robustly on our paired dataset, suggesting that our dataset's quality and alignment significantly aid network training.

From a quantitative standpoint (Table R1), both our GAN-based approach and U-Net achieved high SSIM and PSNR values, with U-Net registering slightly higher scores in those metrics. However, histopathological evaluations rely heavily on the clarity of morphological features. When visually comparing the outputs (Figure R2c–e), we observed that U-Net tends to produce slightly more blurred images, whereas our GAN-based model better preserved intricate subcellular details. This qualitative aspect is particularly crucial for pathology assessments, where the visibility of subtle features can influence diagnostic accuracy.

While conditional diffusion models and contrastive methods have shown promising results in other domains, we note that (1) these techniques can be more computationally demanding and (2) their advantages often become more apparent in unpaired or semi-supervised settings. Given the paired nature of our dataset and the need for high-fidelity structural detail, we found our GAN-based method to offer a favorable balance of image quality, training feasibility, and morphological detail preservation.

To strengthen the credibility of our findings, we have included additional discussion and the comparative results mentioned above in the revised manuscript, providing both quantitative metrics and qualitative assessments. We hope this more comprehensive evaluation highlights the rationale for our chosen architecture and demonstrates that our method is well suited for virtual H&E staining of histological samples.

Figure R2 | Qualitative evaluation of diverse models.

	U-Net	pix2pix	Conditional diffusion	Ours
SSIM	0.8056 \pm 0.0236	0.7580 \pm 0.0216	0.7754 \pm 0.0248	0.7847 \pm 0.0145
PSNR	18.2665 \pm 1.7187	17.3047 \pm 1.8176	11.3599 \pm 1.3261	18.0068 \pm 1.4969

Table R1 | Quantitative evaluation of diverse models.

[2]

How well does the method generalize to previously unseen data? What would happen if the virtual staining would be applied to different tissues?

We appreciate the reviewer's inquiry regarding our model's generalizability. To evaluate how well the model trained on colon tissue images performs on previously unseen data, we applied it to gastric tissue samples (Figure R3). Our findings show that the colon-trained model successfully captured key morphological features—such as nuclei and cytoplasm—even in gastric tissues. However, the resulting virtual H&E images exhibited color discrepancies relative to ground-truth staining, likely due to staining protocol variations between institutions.

To mitigate these differences, we employed the Reinhard color normalization method⁵. After normalization, the model continued to accurately highlight essential features, demonstrating that it can generalize across tissue types. Nonetheless, finer structural details were better preserved when the model was trained specifically on gastric datasets, suggesting that tissue-specific contextual information plays an important role in achieving optimal results.

In summary, our framework can effectively generate foundational morphological features in tissues outside its training domain. Still, for the highest fidelity—particularly for subtle textural details and accurate color representation—training on data from the target tissue type is advised. This tailored training substantially enhances the quality of virtual staining.

Figure R3 | Colon-model applied to the unseen gastric datasets.

[3]

The H&E staining is not directly standardized and the staining of the same tissue sample at different institutions may result in considerably different appearance. How does the proposed method solve the problem? Is there a mechanism to account for the variable concentration of hematoxylin and eosin for “really-stained” samples?

We thank the reviewer for highlighting the significant variability inherent in H&E staining protocols across different institutions. Such variability can arise not only from reagent concentrations but also from factors like staining procedures, reagent manufacturers, and the time elapsed after staining. Numerically accounting for every possible variation is impractical.

Our approach mitigates this challenge by leveraging the quantitative and label-free nature of holotomography, which measures the 3D refractive index (RI) distribution of tissues. Unlike bright-field

images of chemically stained samples, RI values are intrinsic physical properties and remain consistent regardless of staining protocols. Consequently, our method is not affected by fluctuations in hematoxylin or eosin concentrations, nor by color bleaching over extended storage times. Essentially, we bypass the inconsistencies of staining by reconstructing tissue structures from fundamental optical parameters.

Furthermore, as we show in Figure R3, a simple color normalization algorithm (e.g., Reinhard normalization) can effectively align the final virtual H&E colors when transferring models or data between different institutions. Because our virtual staining pipeline reliably captures critical tissue features such as nuclei and cytoplasm, the underlying structural information remains robust. Thus, even when staining protocols differ significantly, our framework can readily adapt via modest color normalization, ensuring consistent and reproducible virtual H&E images.

[4]

Is the holotomography free from slicing artifacts and distortions? How does the virtual staining method behave when it encounters an artifacts caused by the slicing procedure?

We thank the reviewers for highlighting the potential impact of slicing artifacts and distortions. While our holotomography workflow cannot eliminate artifacts that arise during tissue slicing—since they are physical defects introduced prior to imaging—it inherently reduces the slicing steps needed for 50 μm tissue sections by approximately 90% compared to traditional methods. Minimizing the number of slices proportionally decreases the likelihood of artifacts, which are often irreversible once introduced.

Because slicing artifacts were not included in our training dataset, our virtual staining model does not explicitly learn how to correct them. Consequently, any substantial distortions or tears may still appear in the final virtual H&E images as structural anomalies. However, in practice, we have not observed substantial degradation in our reconstructed volumes due to slicing artifacts. Should such artifacts be present, their effect could be partially mitigated by quality-control steps during sample preparation or, in the future, by incorporating artifact-robust training strategies.

[5]

How do the authors ensure that the virtual staining method does not hallucinate non-existing structures?

We appreciate the reviewer's concern regarding potential hallucination of non-existent structures. Our framework minimizes this risk by deriving all structural information directly from holotomography measurements, which capture the 3D refractive index distribution of the tissue. Because the neural network is trained on paired data—where each input holotomography stack corresponds to a chemically stained ground-truth image—its outputs are constrained to reflect only the features present in the physical sample.

Additionally, in Figure 6 of the original manuscript, we show that virtually stained images align closely with the ground truth, reproducing key histological features such as glands and lumens without introducing artificial structures. This consistency indicates that our model does not invent spurious details but instead provides a faithful translation of the inherent tissue morphology. For further validation, we perform expert pathologist reviews to confirm that the model's outputs accurately represent the underlying histopathology.

[6]

The distribution of the SSIM is mostly between 0.75 and 0.82. Is it enough for the downstream tasks? How will the virtually stained 3-D volume work for AI-supported downstream tasks that were trained using the “traditional” 3-D reconstruction methods based on slicing and reconstruction based on image registration? Would it require to retrain or fine-tune already existing architectures to the virtually stained 3-D volumes?

We appreciate the reviewer's question about the adequacy of SSIM values in the range of 0.75 to 0.82 and how our virtual staining approach integrates with downstream AI workflows. First, it is important to note that many downstream tasks—such as segmentation or classification—are typically trained on 2D H&E images. Because our framework generates a virtually stained volume slice by slice, each slice can be treated as an individual 2D H&E image, ensuring compatibility with existing 2D-based pipelines without requiring significant retraining or architectural changes.

Regarding the SSIM values, while they are not at the extreme upper end of the metric, we have demonstrated the practical viability of our virtual H&E images by applying HoverNet⁶, a neural network designed for nuclei segmentation. Our results (Figures 2g–i in the original manuscript) show no statistically significant differences ($p > 0.05$) in the size and number of segmented nuclei between real and virtual H&E images. This indicates that SSIM alone does not fully capture our model's usefulness in downstream applications.

As for 3D tasks that rely on traditionally reconstructed volumes (e.g., stacking multiple 2D slices registered over different depths), retraining or fine-tuning may be beneficial if those architectures were designed with specific image characteristics or distortions in mind. However, our slice-by-slice approach already produces data in a format similar to standard 2D H&E slices, which can be stacked into a 3D volume if needed. In most cases, any adjustments would be modest—such as minor refinements to account for potential domain shifts or differences in image quality—and would not require a complete overhaul of established networks.

[7]

Can the authors quantitatively estimate the benefits of the proposed method? How much does it reduce

the costs and time associated with the 3-D reconstruction compared to the traditional methods?

We thank the reviewer for the opportunity to quantify and further clarify the benefits of our framework. For 3D tissue reconstruction at a thickness of 50 μm , our method can reduce the cost and time associated with slicing and staining by up to 90% compared to conventional approaches. Specifically, traditional histology would require approximately ten separate, thin (e.g., 5 μm) slices for each 50 μm of tissue, each necessitating individual preparation and staining. By contrast, our framework relies on a single tissue section without the need for repetitive chemical staining, drastically decreasing both labor and reagent usage.

Beyond these direct reductions in cost and processing time, the following advantages—referenced in our previous discussions—further underscore the value of our approach:

- **Minimization of artifacts:** Because the tissue is not repeatedly sectioned, the risk of slicing artifacts and misalignment during reconstruction is substantially reduced.
- **Intrinsic image consistency:** Holotomography measures the refractive index distribution, bypassing variations introduced by staining protocols, fixatives, and color bleaching, which can degrade image quality in multi-step procedures.
- **Robust for downstream tasks:** As shown in our HoverNet nuclei segmentation experiments, virtual H&E images produce comparable results to standard H&E slides, supporting a seamless transition to common segmentation or classification pipelines.
- **Scalability to larger samples:** Although high-resolution holotomography can be time-intensive for large volumes, it can be parallelized or optimized via recent advancements in imaging schemes and data processing. This creates a clear path toward high-throughput 3D tissue analysis without the burdensome manual slicing and staining steps required by traditional methods.

In summary, our method substantially lowers operational costs and labor by reducing repetitive slicing and staining steps, while also mitigating several technical artifacts. These benefits collectively enable more efficient and reliable 3D tissue analyses in both research and clinical settings.

[8]

How have the authors ensured that the registration used to create the ground-truth paired volumes is correct? The applied registration method seems like a relatively basic and error-prone method. What about using VALIS, RegWSI, or HistokatFusion?

We thank the reviewer for highlighting the importance of accurate volume registration and for suggesting alternative methods such as VALIS, RegWSI, and HistokatFusion. To assess whether our registration might be error-prone, we conducted a comparative study with RegWSI⁷ (see Figure R4), which demonstrated that our initial approach achieves registration accuracy comparable to RegWSI.

However, we noted that RegWSI reduced overall registration time by approximately 23%, suggesting it could be a more efficient choice.

Beyond this quantitative comparison, we also performed qualitative inspections of critical histological landmarks—such as gland boundaries and nuclear alignments—to ensure that the paired volumes closely matched. These inspections, conducted by experienced pathologists, confirmed that our registrations produced consistent morphological alignment without obvious spatial distortions. While we have not yet tested VALIS or HistokatFusion, we acknowledge they could provide additional benefits in speed or accuracy and plan to explore these methods in future iterations of our workflow.

Figure R4 | Comparing registration methods.

[9]

How can the method be transferred to IHC virtual staining? Does it need to retrain the model for each stain separately? Are there other options? I suggest to introduce a brief description of that in the article.

We thank the reviewer for this insightful question. We believe our 3D virtual staining framework can be extended to immunohistochemistry or immunofluorescence (IHC/IF) by constructing appropriately paired datasets—holotomography paired with corresponding IHC/IF images—and then retraining the model on these data. However, some modifications would be required to accommodate the distinct nature of IHC/IF signals, particularly because they often exhibit lower signal-to-noise ratios and rely on different evaluation criteria (e.g., positive versus negative staining rather than pixel-wise color accuracy).

Additionally, recent advancements in virtual staining have demonstrated techniques for converting H&E images into multiplexed IHC images⁸. By leveraging such methods, it may be possible to generate virtual IHC/IF images directly from our virtual H&E outputs, thereby eliminating the need to retrain a separate model for each immunostain—though supplementary data would likely be necessary to

validate performance and ensure accurate antigen-specific labeling.

To clarify these potential pathways for extending our approach, we have added a discussion in the revised manuscript, outlining how these adaptations or complementary techniques could facilitate virtual IHC/IF staining based on our current framework.

Reviewer #3 (Remarks on code availability):

[10]

The code is very basic, however, it should allow to reproduce the results (by performing the inference on the exemplary cases). The model and exemplary data are released, however, the training cannot be reproduced without releasing the full training dataset.

We have uploaded the training datasets to GitHub and included detailed instructions to facilitate the training process.

Reviewer #4 (Remarks to the Author): Expert in colorectal cancer pathology and molecular pathology

[1]

integration of various growth patterns e.g., mucinous, signet ring cell? Network trained on one 4um thick colon cancer slide. How well is different differentiation patterns represented in virtual HE? integration of normal mucosa commonly found on whole slide sections? how well does the method represent necrosis or apoptotic regions, regions of very crowded nuclei?

We appreciate the reviewer's interest in how our method accommodates various pathological growth patterns, including mucinous and signet ring cell adenocarcinomas, different levels of differentiation, and areas of necrosis or high nuclear density. To examine these concerns, we evaluated our model on an additional tissue slide containing signet ring cells, mucinous adenocarcinoma, normal mucosa, and necrotic regions (Figure R5). Importantly, this slide was processed using the same protocol and trained model described in the manuscript, which was initially developed using 13 tumor microarrays (3 mm in diameter) sourced from different patients.

Our findings indicate that the model effectively captures critical morphological features across diverse pathological contexts. Despite limited training data for signet ring cells and mucinous adenocarcinoma, the virtual H&E images accurately depicted the distinctive nuclear displacement in signet ring cells and clearly distinguished mucin-filled regions. Similarly, normal mucosal tissue, necrotic areas, and zones with high nuclear density were also well-represented, suggesting that the model generalizes robustly beyond its original training distribution.

These results underscore our framework's capacity to address varied tissue architectures and differentiation patterns. We have expanded our discussion in the revised manuscript to highlight this adaptability and to acknowledge that future work could further optimize performance by incorporating an even broader range of tissue types and pathological variations into the training dataset.

Figure R5 | Virtual H&E images of various growth patterns

[2]

how does the method deal with samples prepared in different fixation methods from different institutions or how does age impact applicability of the technology?

We appreciate the reviewer's questions regarding the influence of varying fixation methods and patient age on our approach. While our study focused on FFPE samples using 10% neutral buffered formalin, other common fixatives—such as 4% paraformaldehyde, ethanol, or methanol—are generally reported to affect color or tissue morphology to a lesser extent, rather than induce severe structural distortions⁹⁻¹¹. Since our computational staining relies on the intrinsic refractive index (RI) distribution of label-free tissues, it remains robust as long as the fixative does not induce substantial morphological changes or extreme RI shifts. Although we have not yet tested these alternative fixatives, we consider this an important avenue for future work, as noted in the revised manuscript.

Regarding patient age, we have not observed any age-specific artifacts in our data, and it was not a primary variable in our study. Ultimately, tissue morphology—rather than the patient's chronological age—is more likely to influence the accuracy of our method. Nonetheless, further investigations could examine whether tissue from older patients shows additional artifacts (e.g., increased calcification or fibrosis) that might require modifications to the imaging and staining pipeline.

[3]

i agree one of the primary purposes of H&E is to facilitate visualisation of nuclei and cytoplasm but also nuance of sub-nuclear features such as chromatin intensity. how did these features compare between chemical and virtual?

We thank the reviewer for emphasizing the importance of accurately replicating sub-nuclear features such as chromatin intensity, as these nuances play a crucial role in histopathological assessments. To address this, we conducted a comparative analysis of chromatin intensity and other nuclear details (e.g., nuclear contours) between chemically stained and virtually stained images (Figure R5). Our findings show that the virtual staining method closely mirrors the sub-nuclear structures observed in real H&E images, including chromatin distribution and intensity levels. This alignment is further corroborated by pathologist evaluations, indicating that the virtual H&E provides sufficient fidelity for diagnostic interpretation. We have included a discussion of these observations in the revised manuscript.

[4]

The authors suggest no significant difference between the number and size of nuclei between chemical

and virtual however a trend is observed for average nucleus number ($p=0.0814$). This is likely insufficient for clinical application of such technology. There is a wider distribution of nuclei between patches for virtual compared to chemical - were there any specific features in those with larger discordance between chemical and virtual?

We thank the reviewer for highlighting the importance of precisely quantifying nuclei number and size to ensure clinical viability. In our additional analysis (Figure R6), we compared nuclei counts between chemical and virtual H&E images:

1. Regions with fewer nuclei in virtual H&E (Figures R6c)
 - In these patches, we identified highly elongated nuclei that were partially “suppressed” in the refractive index (RI) data due to surrounding cytoplasmic signals.
 - The all-in-focus algorithm we currently employ emphasizes high-contrast regions, which can overshadow elongated nuclei in areas with stronger cytoplasmic signals.
 - To mitigate this limitation, we plan to refine our axial integration approach. Extending the network architecture to handle 3D RI data more comprehensively (e.g., incorporating adjacent axial slices) could recover these elongated nuclei more accurately.
2. Regions with more nuclei in virtual H&E (Figure R6d)
 - In these instances, segmentation errors led to some nuclei being identified in the virtual images but missed in the chemically stained images.
 - This discrepancy appears to stem from the downstream segmentation algorithm rather than from the virtual staining process itself.

While a statistical trend ($p = 0.0814$) indicates areas for improvement, we believe these discrepancies can be addressed by refining both the axial integration and segmentation pipelines. Moreover, our findings do not negate the clinical feasibility of virtual H&E. Most tissue regions showed close agreement, demonstrating the technique’s potential. Ongoing efforts will focus on improving these edge cases—particularly elongated nuclei detection and segmentation robustness—to further enhance accuracy and facilitate a smooth transition into clinical workflows.

Figure R6 | Comparing nuclei counts between virtual and chemical H&E images

[5]

minor comment - in fig 4d i would move the yellow arrow to point at the lumen - currently it sits on top of the epithelium.

We appreciate the reviewer's suggestion. In response, we have moved the arrows to highlight the lumen and changed their color to ensure better visibility.

[6]

line 280 - "In our direct comparison, we observed circular structures of the nucleus in the input RI images, which is consistent with the predicted H&E images of blue circular morphologies" - these comparisons are not made between virtual method and ground truth of chemical staining - it would be nice to include the canonical chemical stain comparison

We thank the reviewer for the insightful suggestion to improve the clarity of our work. We agree that including real chemical stain images provides a more comprehensive comparison. To address this, we have added the corresponding chemical stain images in Figures 5c and 5d of the revised manuscript.

[7]

line 299 - did the authors consider serial reconstruction of the HE for a more representative "tangible"

comparison between virtual 3D and chemical 3D?

We thank the reviewer for suggesting a more direct comparison between virtual 3D and chemical 3D staining through serial reconstruction of thick tissue sections. While we acknowledge that such an approach could yield valuable insights, it is technically challenging to implement. Once the tissue is deparaffinized and sliced, attempting to re-paraffinize and re-slice the same sample can introduce substantial structural distortions, often resulting in irreversible damage. This damage would compromise the reliability of any subsequent serial reconstruction.

Given these limitations, our current methodology offers the most feasible and accurate comparison under existing constraints. Rather than risking tissue integrity through multiple rounds of slicing and staining, we focus on a direct evaluation of our virtual 3D approach, which avoids repeated manipulation of the same tissue block. We believe this strategy provides a more reliable assessment of our method's performance while minimizing artifacts introduced by additional processing steps.

[8]

Line 315 - comparing 20um thick HE with virtual staining - the authors comment on the chemical staining becoming abnormal at 20um, does this represent a fair comparison of the virtual to the ground truth? a significantly higher cellularity can be observed in fig 6c compared to 6a.

We thank the reviewer for questioning whether comparing a 20 μm chemically stained sample to a virtually stained counterpart is entirely fair, given that chemical staining can become less reliable in thicker tissues. As we discussed in our response to question #7, practical constraints (such as the irreversible structural damage caused by re-slicing) prevent a straightforward serial reconstruction for an ideal ground truth comparison.

Regarding the difference in cellularity observed between Figures 6a and 6c, it largely stems from the way each image is generated. Figure 6a shows a single focal plane—representing only a fraction of the total cellular population—whereas Figure 6c is a composite image that integrates information across the entire 20 μm tissue section. Consequently, Figure 6c naturally appears to have higher cellular density, since it includes cells from multiple focal planes in a single projection.

While this discrepancy may initially seem to undermine a direct comparison, it actually reflects a fundamental advantage of our virtual staining approach: by capturing volumetric information, it provides a more comprehensive view of tissue morphology than any single focal plane can offer in traditional histology.

[9]

what is the scalability of this method? From a clinical utility perspective - number of samples per run,

time per scan?

We appreciate the reviewer's questions regarding the scalability of our approach and its practical utility in a clinical setting. Currently, for a single field of view measuring $220\ \mu\text{m} \times 220\ \mu\text{m} \times 50\ \mu\text{m}$, our method requires about two minutes for holotomography imaging and an additional ten seconds of GPU computation (using an A6000 GPU) to generate the corresponding virtual H&E image.

When extended to a whole-slide scan, however, the total imaging time grows considerably, as each tissue slide must be imaged tile by tile at high resolution. In our setup, this can take up to one week for a full 3D reconstruction of a large tissue section. By contrast, virtual staining can be performed in parallel on multiple GPU nodes—using, for example, four A6000 GPUs (each with 48 GB of memory)—allowing us to produce 3D virtual H&E images for the entire slide within about one day.

We recognize that these time scales may be challenging in clinical workflows, and our discussion in the revised manuscript addresses various strategies to accelerate data acquisition. Ongoing hardware and software advancements—such as faster cameras, high-speed motorized stages, and improved parallelization—are likely to reduce both acquisition and processing times. As these technologies mature, we expect our framework to become increasingly feasible for high-throughput clinical applications, enabling faster imaging and delivering comprehensive 3D tissue insights in a more efficient manner.

[10]

the authors discuss the potential for this technology to overcome current limitations in histopath and immunology to aid in diagnosis and precision medicine - it would be nice to add some sort of clinical comparison e.g., use case to elaborate on this. For example: How does the application of the 3D imaging improve detection of tumour in lymph nodes, if applied to a small clinical cohort what added value does this visualisation method bring for aiding staging? Or quantitation of tumour infiltration lymphocytes?

We appreciate the reviewer's suggestion to illustrate how our 3D imaging framework could enhance clinical utility and precision medicine. One of its core advantages lies in its comprehensive volumetric visualization, which minimizes the risk of missing critical features—such as tumor nodules or tumor-infiltrating lymphocytes (TILs)—that may only be visible in adjacent sections using traditional thin-slice histopathology. By capturing the entire tissue volume in one integrated dataset, our approach can reveal microscopic foci of tumor cells or TILs that otherwise might remain undetected.

In a clinical context, this improved detection sensitivity could be especially impactful for assessing tumor involvement in lymph nodes, aiding more accurate staging. Additionally, the 3D information offers precise delineation of tumor margins, enabling better determination of invasiveness or proximity to resection boundaries. As we refine our method to handle thicker tissue sections and incorporate quantitative analyses, clinicians could leverage automated metrics—such as volumetric TIL density or

tumor volume—to guide treatment planning and prognostication.

Overall, these capabilities point to the transformative potential of our 3D framework in both diagnostic and research settings. By providing more complete visual and quantitative information, our method could not only enhance diagnostic accuracy but also support precision medicine initiatives through detailed assessments of tissue architecture and cellular composition. We have incorporated a more in-depth discussion on these clinical implications into the revised manuscript.

References

- 1 Kiemen, A. L. *et al.* CODA: quantitative 3D reconstruction of large tissues at cellular resolution. *Nature Methods* **19**, 1490-1499 (2022).
- 2 Olson, E., Levene, M. J. & Torres, R. Multiphoton microscopy with clearing for three dimensional histology of kidney biopsies. *Biomed. Opt. Express* **7**, 3089-3096, doi:10.1364/BOE.7.003089 (2016).
- 3 Paul, D., Cowan, A. E., Ge, S. & Pachter, J. S. Novel 3D analysis of Claudin-5 reveals significant endothelial heterogeneity among CNS microvessels. *Microvascular Research* **86**, 1-10, doi:<https://doi.org/10.1016/j.mvr.2012.12.001> (2013).
- 4 Glaser, A. K. *et al.* Light-sheet microscopy for slide-free non-destructive pathology of large clinical specimens. *Nature Biomedical Engineering* **1**, 0084, doi:10.1038/s41551-017-0084 (2017).
- 5 Reinhard, E., Adhikhmin, M., Gooch, B. & Shirley, P. Color transfer between images. *IEEE Computer Graphics and Applications* **21**, 34-41, doi:10.1109/38.946629 (2001).
- 6 Graham, S. *et al.* Hover-net: Simultaneous segmentation and classification of nuclei in multi-tissue histology images. *Medical image analysis* **58**, 101563 (2019).
- 7 Wodzinski, M., Marini, N., Atzori, M. & Müller, H. RegWSI: Whole slide image registration using combined deep feature- and intensity-based methods: Winner of the ACROBAT 2023 challenge. *Computer Methods and Programs in Biomedicine* **250**, 108187, doi:<https://doi.org/10.1016/j.cmpb.2024.108187> (2024).
- 8 Pati, P. *et al.* Accelerating histopathology workflows with generative AI-based virtually multiplexed tumour profiling. *Nature Machine Intelligence* **6**, 1077-1093, doi:10.1038/s42256-024-00889-5 (2024).
- 9 Milcheva, R., Janega, P., Celec, P., Russev, R. & Babál, P. Alcohol based fixatives provide excellent tissue morphology, protein immunoreactivity and RNA integrity in paraffin embedded tissue specimens. *Acta Histochemica* **115**, 279-289, doi:<https://doi.org/10.1016/j.acthis.2012.08.002> (2013).
- 10 Perry, C. *et al.* A Buffered Alcohol-Based Fixative for Histomorphologic and Molecular Applications. *Journal of Histochemistry & Cytochemistry* **64**, 425-440, doi:10.1369/0022155416649579 (2016).

- 11 Snyder, J. M. *et al.* Perfusion with 10% neutral-buffered formalin is equivalent to 4% paraformaldehyde for histopathology and immunohistochemistry in a mouse model of experimental autoimmune encephalomyelitis. *Veterinary Pathology* **59**, 498-505, doi:10.1177/03009858221075588 (2022).